# Projected health and economic impacts of sugar-sweetened beverage taxation in Germany: A cross-validation modelling study

Karl M. F. Emmert-Fees[1,2,3]*, Ben Amies-Cull[4,5], Nina Wawro[3], Jakob Linseisen[6], Matthias Staudigel[7], Annette Peters[3], Linda J. Cobiac[8], Martin O'Flaherty[9], Peter Scarborough[4,5‡], Chris Kypridemos[9‡], Michael Laxy[1,3,10‡]

1 Professorship of Public Health and Prevention, School of Medicine and Health, Technical University of Munich, Munich, Germany, 2 Institute for Medical Information Processing, Biometry, and Epidemiology, Pettenkofer School of Public Health LMU Munich, Munich, Germany, 3 Institute of Epidemiology, Helmholtz Zentrum München, Research Center for Environmental Health, Neuherberg, Germany, 4 Nuffield Department of Primary Care Health Sciences, University of Oxford, Oxford, United Kingdom, 5 Oxford Health Biomedical Research Centre, National Institute of Health and Care Research, Oxford, United Kingdom, 6 Epidemiology, University of Augsburg, University Hospital Augsburg, Augsburg, Germany, 7 TUM School of Management, Technical University of Munich, Munich, Germany, 8 School of Medicine and Dentistry, Griffith University, Southport, Australia, 9 Department of Public Health, Policy & Systems, University of Liverpool, Liverpool, United Kingdom, 10 German Center for Diabetes Research (DZD), Neuherberg, Germany

‡ CK and ML contributed equally and are shared last authors.
* karl.emmert-fees@tum.de

**Data Availability Statement:** The source code for the microsimulation model (IMPACTNCD Germany) used in this study is available in an

## Abstract

### Background

Taxes on sugar-sweetened beverages (SSBs) have been implemented globally to reduce the burden of cardiometabolic diseases by disincentivizing consumption through increased prices (e.g., 1 peso/litre tax in Mexico) or incentivizing industry reformulation to reduce SSB sugar content (e.g., tiered structure of the United Kingdom [UK] Soft Drinks Industry Levy [SDIL]). In Germany, where no tax on SSBs is enacted, the health and economic impact of SSB taxation using the experience from internationally implemented tax designs has not been evaluated. The objective of this study was to estimate the health and economic impact of national SSBs taxation scenarios in Germany.

### Methods and findings

In this modelling study, we evaluated a 20% ad valorem SSB tax with/without taxation of fruit juice (based on implemented SSB taxes and recommendations) and a tiered tax (based on the UK SDIL) in the German adult population aged 30 to 90 years from 2023 to 2043. We developed a microsimulation model (IMPACT_NCD Germany) that captures the demographics, risk factor profile and epidemiology of type 2 diabetes, coronary heart disease (CHD) and stroke in the German population using the best available evidence and national data. For each scenario, we estimated changes in sugar consumption and associated weight change. Resulting cases of cardiometabolic disease prevented/postponed and related quality-adjusted life years (QALYs) and economic impacts from healthcare (medical costs) and

online repository at: https://github.com/kalleEF/IMPACT-NCD-Germany. The used demographic data (population size and composition including projections) can be freely accessed from the GENESIS database of the German Federal Statistical Office at https://www-genesis.destatis.de/genesis/online. Population size and composition are in Table 12411-0013; population projections are in Table 12421-0004. The used mortality data can be freely accessed from the German Federal Health Monitoring System at https://www.gbe-bund.de/gbe/ (Table: "Sterbefälle, Sterbeziffern (ab 1980) according to European Shortlist of Deaths"). The used anthropometric and nutritional exposure data is not freely available and is owned by the respective institutions (Helmholtz Zentrum München for the KORA cohorts; Max Rubner-Institut for the NVS II study). Access to the KORA data is possible upon application via https://helmholtz-muenchen.managed-otrs.com/external/ (contact: KORA.PASST@helmholtz-muenchen.de) and access to the scientific use file of the NVS II study is possible by application to SUF.NVS@mri.bund.de (further information at https://www.mri.bund.de/de/institute/ernaehrungsverhalten/forschungsprojekte/nvsii/scientific-use-file/). All other data is directly accessible in the Supporting Information S1 Appendix or via the used references.

**Funding:** This work was primarily funded through core institutional funding (KEF, NW, AP) from the Helmholtz Zentrum München (German Research Center for Environmental Health, Neuherberg, Germany, https://www.helmholtz-munich.de/) which is funded by the German Federal Ministry of Education and Research (BMBF, https://www.bmbf.de/) and the State of Bavaria. Further, this work was partially funded through the EU Joint Programming Initiative – A Healthy Diet for a Healthy Life (HDHL) Policy Evaluation Network (PEN) project (ML received BMBF grant 01EA1818H). The funders had no role in study design, data collection and analysis, decision to publish, or preparation of the manuscript.

**Competing interests:** KMFEF, BAC, NW, JL, MS, AP, LJC, MOF, CK, and ML report no support from any organisation for the submitted work. PS reports funding support from the National Institute for Health and Care Research (NIHR) Oxford Health Biomedical Research Centre (BRC); BAC reports funding support from the NIHR, Medical Research Council (MRC), National Health Service (NHS) and Wellcome Trust; BAC reports consulting fees from NESTA and UNICEF; BAC reports internal committee positions in the Nuffield Department of Population Health, University of Oxford and

societal (medical, patient time, and productivity costs) perspectives were estimated using national cost and health utility data. Additionally, we assessed structural uncertainty regarding direct, body mass index (BMI)-independent cardiometabolic effects of SSBs and cross-validated results with an independently developed cohort model (PRIMEtime). We found that SSB taxation could reduce sugar intake in the German adult population by 1 g/day (95%-uncertainty interval [0.05, 1.65]) for a 20% ad valorem tax on SSBs leading to reduced consumption through increased prices (pass-through of 82%) and 2.34 g/day (95%-UI [2.32, 2.36]) for a tiered tax on SSBs leading to 30% reduction in SSB sugar content via reformulation. Through reductions in obesity, type 2 diabetes, and cardiovascular disease (CVD), 106,000 (95%-UI [57,200, 153,200]) QALYs could be gained with a 20% ad valorem tax and 192,300 (95%-UI [130,100, 254,200]) QALYs with a tiered tax. Respectively, €9.6 billion (95%-UI [4.7, 15.3]) and €16.0 billion (95%-UI [8.1, 25.5]) costs could be saved from a societal perspective over 20 years. Impacts of the 20% ad valorem tax were larger when additionally taxing fruit juice (252,400 QALYs gained, 95%-UI [176,700, 325,800]; €11.8 billion costs saved, 95%-UI [€6.7, €17.9]), but impacts of all scenarios were reduced when excluding direct health effects of SSBs. Cross-validation with PRIMEtime showed similar results. Limitations include remaining uncertainties in the economic and epidemiological evidence and a lack of product-level data.

## Conclusions

In this study, we found that SSB taxation in Germany could help to reduce the national burden of noncommunicable diseases and save a substantial amount of societal costs. A tiered tax designed to incentivize reformulation of SSBs towards less sugar might have a larger population-level health and economic impact than an ad valorem tax that incentivizes consumer behaviour change only through increased prices.

## Author summary

### Why was this study done?

- Taxation of sugar-sweetened beverages (SSBs), recommended by the World Health Organization (WHO) and implemented in many jurisdictions globally, aims to reduce the noncommunicable disease burden by disincentivizing consumption through increased consumer prices or incentivizing industry reformulation to reduce SSB sugar content.

- No tax on SSBs is currently enacted in Germany and the national government is preparing a new national strategy on food seeking evidence-based recommendations to establish policy priorities until 2050.

- In Germany, the potential long-term health and economic impacts of SSB taxation have not been evaluated.

membership in the Doctoral Training Partnerships Representatives of the MRC; KMFEF, NW, MS, AP, LJC, MOF, PS, CK and ML report no financial relationships with any organisations that might have an interest in the submitted work in the previous three years; JL reports membership in the scientific board and lead of working groups of the German Nutrition Society (DGE) and membership in the Scientific Advisory Board of the Federal Ministry of Food and Agriculture (BMEL); all authors report no other relationships or activities that could appear to have influenced the submitted work.

**Abbreviations:** BMI, body mass index; CHD, coronary heart disease; CVD, cardiovascular disease; GAMLSS, generalised additive models for location, shape and scale; NCD, noncommunicable disease; PUA, probabilistic uncertainty analysis; QALY, quality-adjusted life year; SDIL, Soft Drinks Industry Levy; SSB, sugar-sweetened beverage; T2DM, type 2 diabetes mellitus; UI, uncertainty interval; UK, United Kingdom; US, United States; WHO, World Health Organization.

## What did the researchers do and find?

- We developed and validated a microsimulation model based on national data and international evidence to model the impact of SSB taxation on dietary exposure of added sugar from beverages, body mass index (BMI), cardiometabolic diseases, and related economic costs.

- We evaluated 3 SSB taxation scenarios in Germany with the simulation model: (1) 20% ad valorem tax on SSBs; (2) extended 20% ad valorem tax on SSBs and fruit juice; (3) tiered tax leading to reformulation of SSBs towards 30% lower sugar content.

- Taxation of SSBs in Germany could prevent or postpone 132,100 to 244,100 cases of type 2 diabetes, gain 106,000 to 192,300 quality-adjusted life years (QALYs) and save €10.8 to €16.0 billion in societal cost from 2023 to 2043 with the highest impacts estimated for tiered taxation.

- The absolute long-term health impacts are largely dependent on the relevance of direct, BMI-independent cardiometabolic effects of SSBs.

## What do these findings mean?

- All modelled SSB taxation scenarios are likely to improve population health and reduce societal costs in Germany by preventing cardiometabolic disease.

- Considering all sources of uncertainty, we find that modelled SSB taxation scenarios that lead to reformulation towards less sugar might have a larger population-level health and economic impact than those that incentivize consumer behaviour change only through increased prices.

- From a public health perspective, taxation of SSBs should be considered as a policy option for German decision-makers to reduce consumption of added sugar and improve population health.

## 1. Introduction

In Central Europe, around 27% of premature deaths in 2017 were associated with dietary risk factors [1]. This is assumed to be a direct consequence of food environments fuelling unhealthy diets characterised by a high intake of energy-dense (often ultra-processed) foods high in fat, salt, and sugar [2]. Sugar-sweetened beverages (SSBs), usually defined as beverages with added caloric sweeteners (mostly high-fructose corn syrup and sucrose), are the main source of added sugars in global diets [3]. Excessive consumption of added sugars has been shown to increase morbidity and mortality, indirectly through excess calorie intake leading to weight gain and directly by increasing the risk for coronary heart disease (CHD), type 2 diabetes mellitus (T2DM), and dental caries [3].

To address the health and socioeconomic burden of unhealthy diets, the taxation of unhealthy foods is an important fiscal policy tool with the aim of disincentivizing consumption by increasing prices [4–6]. In recent years, government regulation of SSBs has been gaining traction and over 45 jurisdictions have implemented fiscal policies of which approximately 70% were enacted since 2015 [7]. Additionally, the World Health Organization (WHO) recommends SSB taxation as a best-buy policy to strengthen noncommunicable disease (NCD) prevention [6].

A recent meta-analysis synthesising ex post evaluations of implemented SSB taxation policies indicated that they indeed increase prices, decrease sales and, if designed with a tiered structure, are particularly effective in promoting reductions in added sugars by the food industry (i.e., via reformulation) [4]. A tiered structure here means that tax levels are differentiated based on beverage sugar content using predefined thresholds (e.g., 0.18 British pound sterling [£] per litre for drinks containing 5 to 8 grams [g] sugar per 100 millilitre [ml] and £0.24 per litre for drinks containing more than 8 g sugar per 100 ml in case of the Soft Drinks Industry Levy [SDIL] in the United Kingdom [UK]). However, although early observational studies indicate that SSB taxes could be effective in preventing obesity, generating robust empirical evidence on their impact on long-term health and economic outcomes is difficult [8,9].

In Germany, over 50% of the adult population are overweight and the prevalence of obesity has been increasing to almost 20% over the last years, following a clear socioeconomic gradient [10]. Assessment of the potentially related dietary risk factors is difficult because no regular surveillance of population dietary patterns exists. However, individual studies have shown that most of the population does not follow diet recommendations and that SSB consumption, particularly among younger age groups, is high [11,12].

To improve unhealthy diets, reduce the national burden of NCDs, and increase sustainability of the national food system, the German government is currently gathering recommendations for a national strategy on food [13]. This strategy will cover policy priorities until 2050 in domains such as procurement standards for meals (e.g., as developed by the German Nutrition Society), diet literacy, and sustainable food production [14]. Moreover, while the reduction of sugar in SSBs is a policy objective of the German government, a recent study found that voluntary industry commitments to reduce sugar in soft drinks were not successful [15]. A tax on SSB might be a suitable policy option which is also discussed by German decision-makers and supported by non-governmental organisations such as the German Alliance on NCDs [15].

Some studies have previously attempted to model the health impact of SSB taxation scenarios in Germany. However, these mainly used cohort modelling methods with lower granularity which are not able to account for complex epidemiological dependencies and population dynamics over time and did further not quantify healthcare cost savings or productivity losses [16–18]. The value of these previous studies in providing concrete policy recommendations might thus be limited. Yet, to promote best-practice, cost-effective policies, decision-makers need contemporary and context-specific evidence. This is highlighted by the political processes surrounding the enactment of SSB fiscal policies in Mexico and the UK for which timely scientific evidence played an important role [19].

In this study, we evaluate the impact of context-relevant SSB taxation scenarios in Germany on stroke, CHD, and T2DM morbidity, mortality and healthcare, patient time, and productivity costs with a new individual-level population health microsimulation model (IMPACT$_{NCD}$ Germany). Additionally, we explore structural uncertainty in the predicted policy impact by assessing the relevance of direct, body mass index (BMI)-independent cardiometabolic effects of SSBs and cross-validate our results with a second independently developed Markov cohort simulation model (PRIMEtime) [20–22].

## 2. Methods

### Modelling overview

We evaluated 3 SSB tax policy scenarios that were chosen based on international scientific consensus recommendation and globally implemented SSB taxes accounting for context-relevant factors:

1. 20% ad valorem tax on SSBs based on scientific recommendations and implemented ad valorem or volumetric taxes in, e.g., Mexico, Chile, and United States (US) legislatures [4,6,7,23];

2. 20% ad valorem tax on SSBs and fruit juice, extending the tax to account for high consumption levels and the caloric content of juices [12];

3. 30% reformulation of SSBs towards lower sugar content based on tiered taxes such as the UK SDIL [4,7].

Further details on all scenarios and related assumptions are described below and in Methods A in S1 Appendix under "Policy module."

To simulate the health and economic impact of these SSB taxation scenarios, we developed and validated an NCD microsimulation model for Germany based on the UK IMPACT$_{NCD}$ framework (IMPACT$_{NCD}$ Germany; hereafter IMPACT$_{NCD}$). We modelled the German population age 30 to 90 years over 20 years (2023 to 2043) and performed an economic evaluation from healthcare and societal perspectives [24–27]. For this, we created a synthetic population stratified by age and sex that captures the real demographics, exposures, dietary intakes, and disease epidemiology of the actual German population using available national data sources (see below and in Methods A in S1 Appendix under "Epidemiological engine").

The main pathways of our model are founded on widely accepted epidemiological evidence and summarised in Fig 1. Briefly, in our main analysis we assumed that SSB taxation would, depending on the scenario, induce changes in SSB and fruit juice consumption or SSB sugar content guided by economic theory and observations from other countries in which taxes were implemented. We then modelled the effect of changed sugar consumption from SSBs and fruit juice on BMI and consequently BMI and SSB intake as exposures for T2DM, CHD, and stroke. Direct (BMI-independent) effects of SSBs on T2DM and CHD are assumed to be due to the added sugar they contain [21,28]. We also considered T2DM as a risk factor for CHD, stroke, and non-cardiovascular mortality. To analyse structural uncertainty, we (1) re-estimated all scenarios using only BMI-mediated effects, thus excluding the potentially more uncertain estimates of the direct cardiometabolic effects of SSBs from nutritional

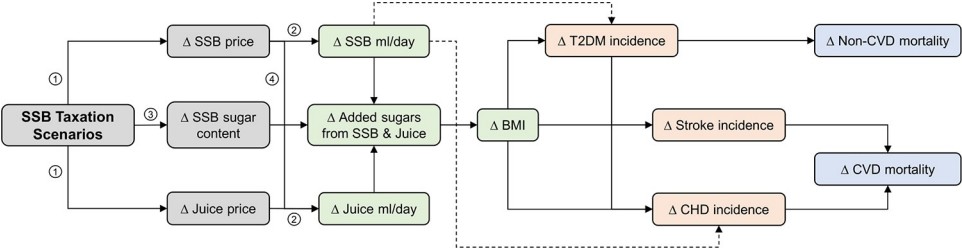

**Fig 1. Policy, exposure, and disease pathways of the SSB tax modelling with IMPACT$_{NCD}$ Germany.** Complete modelled pathways. Dashed arrows indicate direct effects of SSBs that are excluded in structural uncertainty analyses. Grey boxes indicate modelled policy pathways and corresponding numbered arrows denote underlying modelled mechanisms: (1) Economic theory suggests that SSB producers will pass some proportion of a tax on SSBs along to consumers via increased prices (tax pass-through). (2) After accounting for the tax pass-through rate, economic theory suggests that increased prices of SSBs lead to a change in SSB consumption based on their own-price elasticity of demand. (3) Tiered taxation with different tax rates according to SSB sugar content, incentivizes reformulation towards lower sugar content by producers to avoid the tax burden. (4) After accounting for the tax pass-through rate, economic theory suggests that increased prices of SSBs lead to substitution with similar goods such as fruit juice based on their cross-price elasticity of demand. Green boxes indicate exposures. Orange boxes and blue boxes indicate disease and mortality outcomes, respectively. Δ, "change in"; BMI, body-mass index; Juice, fruit juice; CHD, coronary heart disease; CVD, cardiovascular disease; ml, millilitre; SSB, sugar-sweetened beverages; T2DM, type 2 diabetes mellitus.

epidemiological studies [21]; and (2) cross-validated these results with the PRIMEtime cohort model (see below and Methods B and Methods C in S1 Appendix) [29].

Our model choice was guided by a previous review that we conducted during the Policy Evaluation Network project [30]. The technical details of both models have been described previously [20,26,27]. For an overview of all used parameters and data sources see Tables A and G in S1 Appendix. We provide the source code for IMPACT$_{NCD}$ in a public repository at https://github.com/kalleEF/IMPACT-NCD-Germany.

## Sugar-sweetened beverage taxation scenarios

All SSB taxation scenarios including the additional taxation of fruit juice in Germany were modelled compared to a baseline without tax. SSBs were defined as (un-)caffeinated soft drinks or fruit drinks with added sugars (i.e., caloric sweeteners). Fruit juice was defined as 100% fruit juice and nectars or other types of juices that might contain added sugars. Unfortunately, we were not able to distinguish types of juices in detail. However, we consider the resulting bias as negligible because over 80% of consumed fruit juice in Germany does not contain added sugars [31].

The modelled SSB tax policy scenarios were based on globally implemented taxes and international scientific consensus recommendations [6,23]. Three scenarios were selected according to their relevance for the German context and considering limitations of the available beverage consumption data. The modelled scenarios were:

1. 20% ad valorem tax on SSBs (i.e., the tax is calculated as a proportion of the price) based on implemented ad valorem or volumetric taxes as for example in Mexico, Chile, and US legislatures [7]. Due to lack of access to product-level ingredient and price data, we could only approximate volumetric taxes through this scenario. However, most implemented taxes were designed to increase prices by 10% to 20% based on scientific recommendations [6,23]. Evidence has shown that such taxes indeed have increased prices and reduced SSB consumption (hereafter: "ad valorem tax") [4,6,7,23];

2. 20% ad valorem tax on SSBs and fruit juice that extends SSB taxation to account for the caloric content of fruit juices and the high baseline consumption in Germany (hereafter: "extended ad valorem tax");

3. tiered tax design with increasing tax rates according to specific sugar thresholds that incentivizes producers to reformulate SSBs towards 30% lower sugar content. This was based on observed reformulation effects under the UK SDIL, which could serve as a blueprint policy for other European countries such as Germany (hereafter: "tiered tax") [15,32,33].

Further details on the scenarios and their implementation in the model are described below and in Methods A in S1 Appendix under "Policy module."

## Synthetic German population

To simulate the population-level impact of the policy scenarios, we constructed a synthetic German population. For this we combined (1) data on the exposures BMI and intake of SSBs and fruit juice from 3 waves of the Kooperative Gesundheitsforschung in der Region Augsburg (KORA) study (S4, F4, and FF4; 1999, 2007, and 2014), a population-based cohort in southern Germany, and the nationally representative dietary survey Nationale Verzehrsstudie (NVS) II (2006) [34–36]; (2) national data on the epidemiology of stroke, CHD, and T2DM [37]; and (3) information on death counts, population count estimates, and projections by age and sex from official sources and a de novo mortality forecast using a functional demographic model

(Methods A in S1 Appendix under "Mortality calibration") [38–41]. Detailed step-by-step descriptions of all data sources, applied statistical methods, and validation results for the synthetic population are described in Methods A in S1 Appendix under "Epidemiological engine."

## Estimating exposure distributions

To estimate exposure distributions conditional on age and sex, we used generalised additive models for location, shape and scale (GAMLSS), which can handle complex relationships between the response variable and its predictors and numerous types of distributions [42]. For the distribution of BMI, we used data from the 3 KORA waves to incorporate time trends. In KORA FF4 usual dietary intakes were calculated using a blended approach consisting of up to three 24-h food lists and a food frequency questionnaire [35]. In NVS II dietary intakes were calculated based on two 24-h dietary recalls [34]. For the estimation of the SSB and fruit juice intake distributions in ml/day, we primarily relied on KORA FF4 and supplemented this with the NVS II due to the underrepresentation of younger age groups in KORA FF4. Our method accounts for non- and high-consumers in the beverage intake distributions via mixture models. We calculated sugar intake from SSBs and fruit juice using information on beverage-specific sugar consumption in g/ml that was available in KORA FF4. We adjusted beverage and sugar intake for misreporting with the residual method (i.e., regressed on energy intake) [43]. Further details are in Methods A in S1 Appendix under "Exposure module."

## Estimating disease epidemiology

Information on the epidemiology of T2DM was gathered from the most recent data of the German national diabetes surveillance [37]. Due to data limitations for stroke and CHD incidence, we applied SCORE2 risk equations to the KORA data to estimate the yearly incidence of cardiovascular disease by age and sex [44, 45]. Incidence trends were based on the empirical trends between follow-ups S4, F4, and FF4. The proportions of stroke and CHD events in these estimates were based on results from the European Prospective Investigation into Cancer and Nutrition (EPIC) [46]. Finally, all epidemiological data was aligned and smoothed with DISMOD II before application in IMPACT$_{NCD}$ to improve consistency between incidence, prevalence, and mortality data [47]. Further details are in Methods A in S1 Appendix under "Disease module."

## Effect of SSB taxation scenarios on sugar intake

According to economic theory, taxation can have both demand and supply side effects [48]. In the case of SSBs, taxation leads to increased consumer prices as producers pass some proportion of the tax along (i.e., tax pass-through) [48]. Price changes influence consumption of goods due to own- and cross-price elasticities of demand (i.e., %-change in consumption following price increase of the same or another product category by 1%) [49]. However, SSB taxes (e.g., the UK SDIL) can be designed with tax levels that depend on beverage sugar content and thus incentivize product reformulation, giving producers a way to avoid the tax. These mechanisms have important implications on implementing the analysed policy scenarios in our model. For further details, see Methods A in S1 Appendix under "Policy module."

In the "ad valorem tax" and "extended ad valorem tax" scenarios, the effect of price increases on SSB and fruit juice consumption was modelled with national price elasticities of demand. We first calculated the change in consumption of SSB and fruit juice and consequently the change in respective sugar intake. In the former we considered substitution from SSBs to fruit juice. In both scenarios, we assumed that the policy immediately affected beverage

consumption. Because national elasticities were not available, we estimated de novo uncompensated price elasticities for beverage categories with an Almost Ideal Demand System using data from the German household consumption survey (Einkommens- und Verbrauchsstichprobe 2013 and 2018) (Methods A in S1 Appendix under "Policy module"). We estimated the own-price elasticity of SSBs and fruit juice to be −0.956 (95%-confidence interval [−1.174, −0.738], $p < 0.001$) and −1.106 (95%-CI [−1.397, −0.814], $p < 0.001$), respectively. The cross-price elasticity for SSBs and fruit juice was estimated to be 0.052 (95%-CI [−0.138, 0.242], $p = 0.593$) (Table J in S1 Appendix). Based on a recent meta-analysis, we assumed that the tax pass-through was 82% (95%-CI [66,98]; $p < 0.001$) [4].

To implement the effects of reformulation in the "tiered tax" scenario, we had to make several assumptions. Based on a recent evaluation of the UK SDIL, we assumed that individual intake of sugar from SSBs would be reduced by 30% without changing consumption [15]. We additionally assumed that this reduction would be gradually come into effect over 3 years. The reformulation effect can only be approximated since we did not have access to product-level data in the KORA or NVS studies and were thus unable to directly specify the underlying sugar thresholds. This means we assumed that a tiered tax, replicating the design of the UK SDIL and taking German SSB price levels and sugar content into account, would gradually lead to similar reformulation effects as in the UK.

### Effects of exposures on cardiometabolic risk

For all modelled exposure and disease pathways in Fig 1, we used high-quality evidence from meta-analyses of prospective cohort studies or randomised controlled trials and cohort pooling projects to translate changes in sugar and SSB intake into changes in BMI and the risk for stroke, CHD, and T2DM. For simplicity, we considered the metabolic effects of sugar from SSBs and fruit juice on BMI to be the same [50].

We modelled the ceteris paribus effect of reduced consumption of sugar from beverages (i.e., no compensation behaviour beyond considered beverages) on the long-term reduction in BMI with effect estimates from a meta-analysis of prospective cohorts [51]. The predicted reductions in weight are conservative and generally lower than using traditional energy balance equations [52,53] (for comparisons, see Table G in S1 Appendix).

Like previous diet policy modelling studies, we considered (1) BMI-mediated effects on stroke, CHD, and T2DM risk [27,54,55]; (2) BMI-independent (direct) effects of SSBs on CHD and T2DM risk due to their sugar content [21,50,56]; and (3) T2DM as a risk factor for stroke, CHD, and non-cardiovascular mortality [57]. We included direct effects of SSBs to reflect potential underlying mechanisms related to insulin resistance and inflammation but excluded them in structural uncertainty analyses (see below) [21,50,58,59]. However, we did not model potential direct effects of fruit juice due to higher uncertainty in these estimates [21]. An overview of all used risk parameters can be found in Table G in S1 Appendix.

### IMPACT$_{NCD}$ microsimulation

IMPACT$_{NCD}$ is a dynamic, discrete-time, stochastic, open-cohort microsimulation model that simulates the life course of individuals and their counterfactuals under alternative policy scenarios. It enables the detailed simulation of diet policies and their impact on relevant exposures, subsequent disease epidemiology, and mortality in a competing risk framework accounting for different lag-times between exposures and outcomes. The effect of individual exposure changes on disease risk is achieved with individualised attributable fractions (Methods A in S1 Appendix under "Disease module"). Our model was calibrated to observed (2013 to 2019) and future (2020 to 2043) non-cardiovascular (CVD) mortality rates projected with a

functional demographic model [38]. Epidemiological and economic outputs of the model on a population level are highly flexible and aggregated from individual life courses. We mainly report (incident) cases and (prevalent) case-years prevented/postponed (i.e., either completely prevented or delayed by 1 or more years). A detailed technical description of the modelling process is given in Methods A in S1 Appendix.

## Health-related medical, patient time, and productivity costs

We estimated medical, patient time, and productivity costs related to morbidity and mortality over the simulation period based on the life course of synthetic individuals. Economic impacts were calculated from healthcare and societal perspectives following contemporary economic evaluation guidelines using a human capital approach [60,61]. We assessed formal health sector costs by applying medical costs for the treatment of stroke, CHD, and T2DM from the newest available national evidence to each incident or prevalent case year [62,63]. In the societal perspective, we additionally assessed informal health sector (i.e., patient time costs for T2DM self-management and health services use) and productivity costs (i.e., sick leave days and early retirement associated with stroke and T2DM and premature death) [64]. These were valued according to published national estimates, except premature death costs that were calculated using average annual gross wages including fringe benefits [65–68]. Medical and patient time costs were applied until death, while productivity losses accumulated until the German retirement age of 65 years. We inflated health sector costs, informal health sector costs, and productivity losses to 2022 prices using the German medical sector price index and the German labour cost index (see Methods A in S1 Appendix under "Health economics module") [69,70]. All costs were discounted at 3% per year [71].

## Quality-adjusted life years

Cumulative quality-adjusted life years (QALYs) over the simulation period were estimated taking the health-related quality of life of synthetic individuals, including morbidity and mortality, over their life course into account. For this, we re-estimated recently published national health utility decrements accounting for age, sex, BMI, stroke, CHD, and T2DM to improve consistency with our analysis (see Methods A in S1 Appendix under "Health economics module") [72]. QALYs were discounted at 3% per year [71].

## Uncertainty and sensitivity analyses

IMPACT$_{NCD}$ incorporates stochastic (first-order) and parameter (second-order) uncertainty, as well as individual heterogeneity with extensive probabilistic (Monte Carlo) uncertainty analyses (PUA) using 500 iterations [73,74]. We also assessed structural uncertainty in the predicted policy impact with regards to BMI-independent effects of SSBs by re-estimating all scenarios using only BMI-mediated effects (Fig 1) [74].

   We conducted several sensitivity analyses to contextualise our results by providing further comparison scenarios and varying important policy-related assumptions. We modelled: (1) impacts of observed voluntary reformulation of SSBs by industry (i.e., 2% per 6 years) [15]; (2) the tiered tax scenario with reformulation by 10%; (3) tax rates of 10% and 30% for the "ad valorem tax" scenario; (4) the "ad valorem tax" scenario without substitution effects to fruit juice (i.e., setting the cross-price elasticity to 0); (5) the impact of price changes on SSB consumption with a meta-analytic estimate [75]; (6) a maximum impact scenario that combines reformulation and consumption reduction; and (7) varied discount rates for costs and QALYs (0%, 5%, and 10%). Further details on uncertainty and sensitivity analyses are available in Methods A and B in S1 Appendix.

## Model validation

We performed extensive analyses to validate IMPACT$_{NCD}$ according to current guidelines [29]: (1) To ensure internal and face validity, the computational implementation, model outputs, and structure were discussed during meetings among the author group. We also compared inputs and model outputs of disease-specific prevalence and incidence and assessed the ability of the model to track past observed and projected mortality (Fig S–AD in S1 Appendix). (2) To assess external validity, we compared simulated disease-specific epidemiological data from the baseline scenario to comparable external information not used to inform model inputs (Fig AE–AM in S1 Appendix). (3) To cross-validate the predicted policy impact, we modelled all policy scenarios with an adapted version of PRIMEtime, which is a discrete-time proportional multistate life table Markov cohort model. Due to PRIMEtime's preexisting structure, cross-validation was performed for the analysis including only BMI-mediated effects. As cross-validation targets, we used disease-specific cases prevented/postponed and QALYs. We minimised potential differences between models, by generating exposure and epidemiological inputs for PRIMEtime based on the synthetic German population; using the same data sources where possible; and aligning preexisting disease risk parameters. Uncertainty in PRIMEtime was separately assessed with 1,000 PUA iterations and only includes parameter (second-order) uncertainty. Technical descriptions of PRIMEtime have been published previously [20,76]. See Methods B and C in S1 Appendix for an overview of PRIMEtime and an extensive description of the cross-validation.

## Patient and public involvement

No patients or members of the public were involved in the design and conduct of this study or the interpretation of its results. The results of this study will be shared as policy briefs with public representatives. We acknowledge that this analysis would not have been possible without data from participants of the respective cohort and survey studies.

## 3. Results

We estimated that a national tax on SSBs and/or juice would decrease consumption of sugar from these beverages in Germany by 1 g/day (95%-uncertainty interval [UI] [0.05 to 1.65]) in the "ad valorem tax" scenario; 5.91 g/day (95%-UI [5.37, 6.04]) in the "extended ad valorem tax" scenario and 2.34 g/day (95%-UI [2.32, 2.36]) in the "tiered tax" scenario (Table 1). An overview of all exposure changes by scenario is given in Table S in S1 Appendix. Over 20 years, all modelled scenarios would have a positive impact on population health in Germany, reducing obesity, saving healthcare costs, and leading to productivity gains (Table 2 and Fig 2). This finding was robust, irrespective of the simulation model used and whether direct effects of sugar in SSBs were considered, albeit estimated impacts were a lot lower when excluding them (Fig 2). We mainly focused on the results including all exposure pathways in the following sections. See Tables T–Z in S1 Appendix for stratified and additional analyses.

## Health impact of SSB taxation scenarios

A 20% ad valorem tax (scenario: "ad valorem tax") on SSBs could prevent/postpone around 1,900 cases of stroke (95%-UI [0, 4,500]), 39,200 cases of CHD (95%-UI [21,100, 58,100]), 132,100 cases of T2DM (95%-UI [61,700, 202,900]), and 31,600 (95%-UI [−5,400, 72,600]) cases of obesity, compared to the counterfactual without the policy, and 1,109,300 (95%-UI [481,700, 1,838,200]) case-years lived with T2DM and 733,800 (95%-UI [99,600, 1,431,500]) case-years lived with obesity could be mitigated (Table 2).

**Table 1. Changes in sugar consumption from SSBs and juice under different policy scenarios in Germany by sex and age groups.**

| Sex and age group | Change in sugar consumption from SSBs and juice compared to baseline without tax (95%-uncertainty intervals) | | |
|---|---|---|---|
| | Ad valorem tax* | Extended ad valorem tax# | Tiered tax‡ |
| | *Absolute change from 2023 to 2043 in g/day* | | |
| Male | | | |
| 30–49 years | −2.87 (−3.82, −1.36) | −10.19 (−10.44, −9.28) | −6.11 (−6.22, −6.02) |
| 50–69 years | −1.32 (−2.04, −0.24) | −6.81 (−6.99, −6.17) | −2.99 (−3.04, −2.94) |
| 70–90 years | −0.57 (−1.03, 0.12) | −4.09 (−4.21, −3.69) | −1.39 (−1.42, −1.35) |
| Female | | | |
| 30–49 years | −0.90 (−1.69, 0.26) | −6.93 (−7.12, −6.28) | −2.24 (−2.27, −2.20) |
| 50–69 years | −0.44 (−1.06, 0.43) | −5.17 (−5.32, −4.67) | −1.25 (−1.27, −1.23) |
| 70–90 years | −0.22 (−0.64, 0.36) | −3.36 (−3.47, −3.03) | −0.70 (−0.71, −0.68) |
| *Total* | *−1.00 (−1.65, −0.05)* | *−5.91 (−6.04, −5.37)* | *−2.34 (−2.36, −2.32)* |
| | *Relative change from 2023 to 2043 in %* | | |
| Male | | | |
| 30–49 years | −4.73 (−6.35, −2.24) | −16.97 (−17.44, −15.48) | −10.12 (−10.34, −9.94) |
| 50–69 years | −3.55 (−5.45, −0.67) | −18.03 (−18.60, −16.33) | −8.02 (−8.17, −7.86) |
| 70–90 years | −2.53 (−4.57, 0.44) | −17.91 (−18.50, −16.14) | −6.17 (−6.30, −6.02) |
| Female | | | |
| 30–49 years | −2.23 (−4.28, 0.68) | −17.44 (−18.01, −15.84) | −5.60 (−5.71, −5.50) |
| 50–69 years | −1.58 (−3.78, 1.54) | −18.36 (−18.98, −16.57) | −4.46 (−4.54, -4.38) |
| 70–90 years | −1.17 (−3.46, 2.03) | −18.41 (−18.99, −16.51) | −3.71 (−3.79, -3.64) |
| *Total* | *−2.94 (−4.80, −0.18)* | *−17.15 (−17.54, −15.60)* | *−6.85 (−6.92, -6.79)* |

Absolute and relative changes in sugar consumption (in grams per day) from SSBs and juice under different policy scenarios in Germany by sex and age groups.

*"Ad valorem tax" refers to a 20% ad valorem tax on SSBs with a pass-through to consumers of 82% (for details, see section "Sugar-sweetened beverage taxation scenarios").

#"Extended ad valorem tax" refers to a 20% ad valorem tax on SSBs and fruit juice with a pass-through to consumers of 82% (for details, see section "Sugar-sweetened beverage taxation scenarios").

‡"Tiered tax" refers to a tiered tax on SSBs similar to the UK SDIL that leads to a reduction in SSB sugar content by 30% through reformulation (for details, see section "Sugar-sweetened beverage taxation scenarios").

g, grams; SSB, sugar-sweetened beverages; UK SDIL, United Kingdom Soft Drinks Industry Levy.

Expanding the tax to fruit juice (scenario: "extended ad valorem tax") would lead to increased overall health gains (cases prevented/postponed: 4,500 for stroke, 95%-UI [1,900, 8,500]; 45,800 for CHD, 95%-UI [27,500, 66,200]; 190,800 for T2DM, 95%-UI [112,000, 269,700]) (Table 2), and the largest impact on obesity, particularly among women due to their relatively high fruit juice consumption (Methods A in S1 Appendix under "Exposure module"). Obesity cases prevented/postponed would increase to 109,700 (95%-UI [70,400, 162,200]) for men and 50,500 (95%-UI [22,100, 80,900]) for women (Table X in S1 Appendix).

In the "tiered tax" scenario, health impacts for CHD and T2DM were substantially higher than in the "(extended) ad valorem tax" scenarios. Here, the cases prevented/postponed increased to 3,400 for stroke (95%-UI [800, 7,100]), 69,800 for CHD (95%-UI [38,800, 101,900]), 244,100 for T2DM (95%-UI [118,200, 365,300]), and 72,300 (95%-UI [36,400, 105,500]) for obesity (Table 2).

**Table 2. Health and economic impact of different SSB taxation scenarios in Germany 2023–2043 from healthcare and societal perspectives.**

| Health outcomes | Change in outcomes compared to baseline without tax (95%-uncertainty intervals) | | |
|---|---|---|---|
| | Ad valorem tax* | Extended ad valorem tax# | Tiered tax‡ |
| Cases prevented/postponed[†] | | | |
| T2DM | 132,100 (61,700, 202,900) | 190,800 (112,000, 269,700) | 244,100 (118,200, 365,300) |
| CHD | 39,200 (21,100, 58,100) | 45,800 (27,500, 66,200) | 69,800 (38,800, 101,900) |
| Stroke | 1,900 (0, 4,500) | 4,500 (1,900, 8,500) | 3,400 (800, 7,100) |
| Obesity | 31,600 (−5,400, 72,600) | 159,400 (97,100, 232,400) | 72,300 (36,400, 105,500) |
| Case-years prevented/postponed[†] | | | |
| T2DM | 1,109,300 (481,700, 1,838,200) | 1,569,600 (876,500, 2,313,800) | 1,940,900 (879,200, 3,106,500) |
| CHD | 239,700 (112,300, 375,600) | 274,700 (146,600, 415,100) | 408,200 (206,600, 620,900) |
| Stroke | 4,300 (−6,600, 22,200) | 18,600 (2,300, 50,200) | 8,900 (−6,600, 32,300) |
| Obesity | 733,800 (99,600, 1,431,500) | 3,919,200 (2,340,700, 5,490,500) | 1,683,100 (1,035,800, 2,341,000) |
| All-cause deaths prevented/postponed | 17,000 (8,600, 26,100) | 21,600 (12,600, 31,800) | 29,300 (15,900, 44,900) |
| QALYs gained | 106,000 (57,200, 153,200) | 252,400 (176,700, 325,800) | 192,300 (130,100, 254,200) |
| Life years gained | 95,400 (47,300, 161,000) | 114,200 (61,300, 187,300) | 156,700 (77,900, 255,400) |
| Difference in life expectancy | 0.02 (−0.01, 0.05) | 0.02 (−0.01, 0.06) | 0.03 (0.00, 0.08) |
| Difference in life expectancy at age 60 years | 0.01 (−0.01, 0.02) | 0.01 (−0.01, 0.03) | 0.01 (−0.01, 0.04) |
| Health-related cost outcomes (€-millions) | | | |
| Healthcare costs | | | |
| T2DM | −1,613 (−2,750, −684) | −2,310 (−3,492, −1,311) | −2,785 (−4,601, −1,249) |
| CHD | −660 (−1,003, −345) | −792 (−1,170, −461) | −1,136 (−1,631, −619) |
| Stroke | −89 (−225, 0) | −204 (−425, −84) | −153 (−364, −35) |
| Other | 118 (59, 202) | 144 (77, 236) | 190 (97, 326) |
| Productivity costs | | | |
| T2DM early retirement | −24 (−60, 5) | −35 (−73, −3) | −41 (−101, 4) |
| T2DM sick leave | −1,170 (−2,809, −384) | −1,536 (−3,427, -541) | −2,013 (−4,707, −610) |
| Stroke early retirement | −1 (−33, 5) | −5 (−70, 0) | −3 (−62, 3) |
| Stroke sick leave | 0 (−1, 0) | 0 (−3, 0) | 0 (−2, 0) |
| Premature death | −3,556 (−7,135, −1,260) | −3,904 (−7,388, −1,518) | −5,913 (−10,999, −2,265) |
| Time costs | | | |
| T2DM self-management | −1,146 (−2,020, −475) | −1,461 (−2,420, −730) | −1,941 (−3,368, −863) |
| T2DM time for health service use | −1,446 (−3,508, −502) | −1,892 (−4,424, −718) | −2,470 (−5,667, −847) |
| Other time for health service use | 374 (153, 651) | 485 (239, 781) | 633 (269, 1,068) |
| Cost-effectiveness | | | |
| Total change in costs from healthcare perspective (€-millions) | −2,262 (−3,596, −1,189) | −3,141 (−4,568, −1,942) | −3,850 (−6,075, −2,070) |
| Total change in costs from societal perspective (€-millions) | −9,584 (−15,304, −4,714) | −11,827 (−17,887, −6,702) | −16,013 (−25,500, −8,090) |
| ICER[§] (healthcare perspective) | Dominant | Dominant | Dominant |
| ICER[§] (societal perspective) | Dominant | Dominant | Dominant |

*"Ad valorem tax" refers to a 20% ad valorem tax on SSBs with a pass-through to consumers of 82% (for details, see section "Sugar-sweetened beverage taxation scenarios").

#"Extended ad-valorem tax" refers to a 20% ad valorem tax on SSBs and fruit juice with a pass-through to consumers of 82% (for details, see section "Sugar-sweetened beverage taxation scenarios").

‡"Tiered tax" refers to a tiered tax on SSBs similar to the UK SDIL that leads to a reduction in SSB sugar content by 30% through reformulation (for details, see section "Sugar-sweetened beverage taxation scenarios").

[†]Cases and case-years prevented/postponed are defined as incident and prevalent cases completely prevented or delayed for 1 or more years, respectively.

[§]Expressed as € per QALY. Only defined for positive incremental costs and else "dominant" because no trade-off exists if health is improved while costs are saved.

CHD, coronary heart disease; ICER, incremental cost-effectiveness ratio; QALY, quality-adjusted life year; SSB, sugar-sweetened beverages; T2DM, type 2 diabetes mellitus; UK SDIL, United Kingdom Soft Drinks Industry Levy.

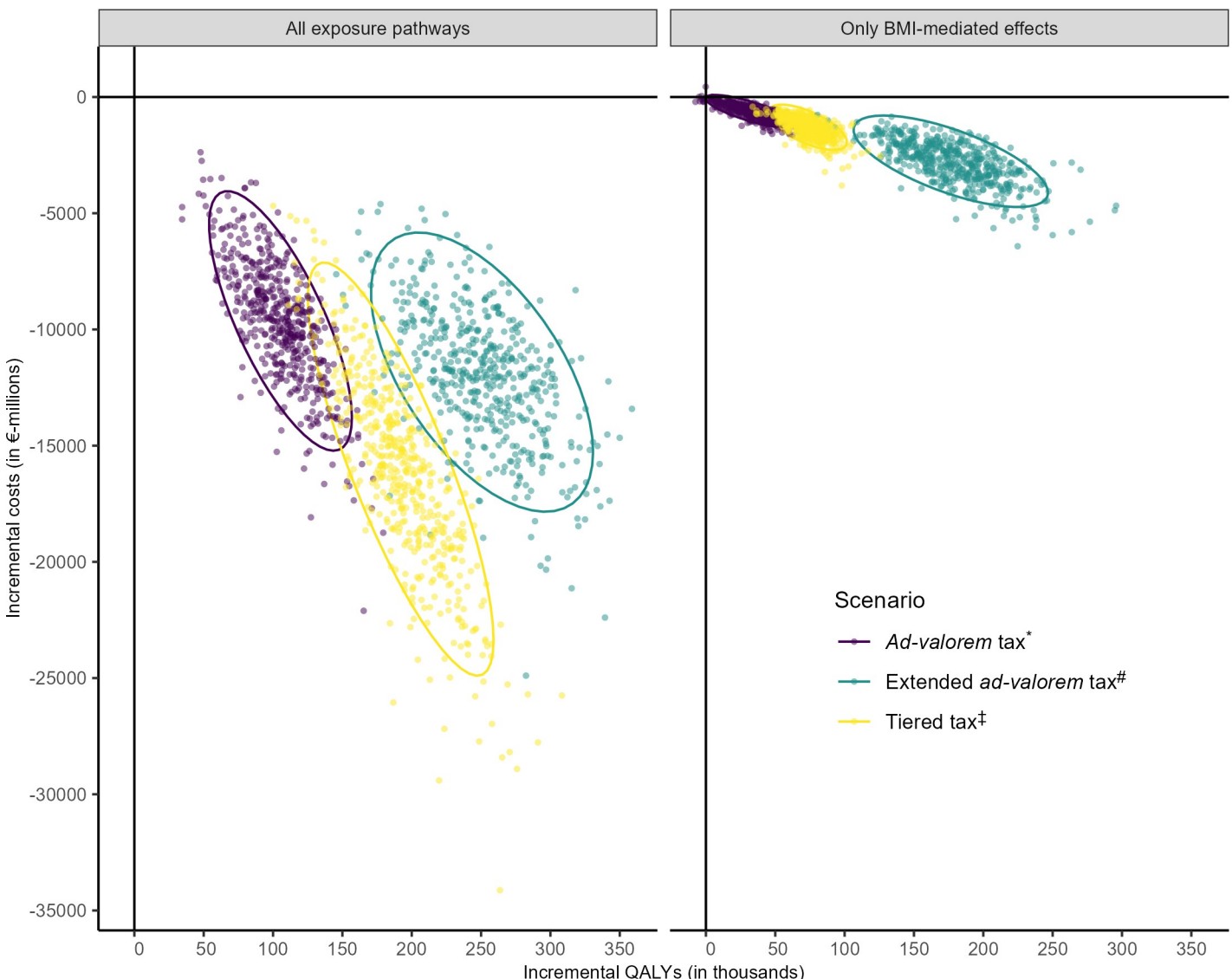

**Fig 2. Cost-effectiveness plane for different SSB taxation scenarios in Germany 2023–2043 from a societal perspective.** Cost-effectiveness plane showing incremental (i.e., compared to baseline without tax) QALYs on the x-axis and incremental costs from a societal perspective on the y-axis by scenario and included exposure pathways. Ellipses indicate 95%-uncertainty intervals of incremental QALYs and costs per scenario assuming a multivariate t-distribution. *"Ad valorem tax" refers to a 20% ad valorem tax on SSBs with a pass-through to consumers of 82% (for details, see section "Sugar-sweetened beverage taxation scenarios"). #"Extended ad valorem tax" refers to a 20% ad valorem tax on SSBs and fruit juice with a pass-through to consumers of 82% (for details, see section "Sugar-sweetened beverage taxation scenarios"). ‡"Tiered tax" refers to a tiered tax on SSBs similar to the UK SDIL that leads to a reduction in SSB sugar content by 30% through reformulation (for details, see section "Sugar-sweetened beverage taxation scenarios"). BMI, body mass index; QALY, quality-adjusted life year; SSB, sugar-sweetened beverages; UK SDIL, United Kingdom Soft Drinks Industry Levy.

For all scenarios, the policy impact was highest among men due to their higher baseline SSB consumption, particularly in younger ages. Most cases of stroke, CHD, and T2DM would be prevented in the age groups below 70 years (Table W in S1 Appendix).

### Economic impact of SSB taxation scenarios

Healthcare costs saved, productivity loss averted and QALYs gained over the 20-year simulation period were substantial in all scenarios and mirror the reported health impacts in their

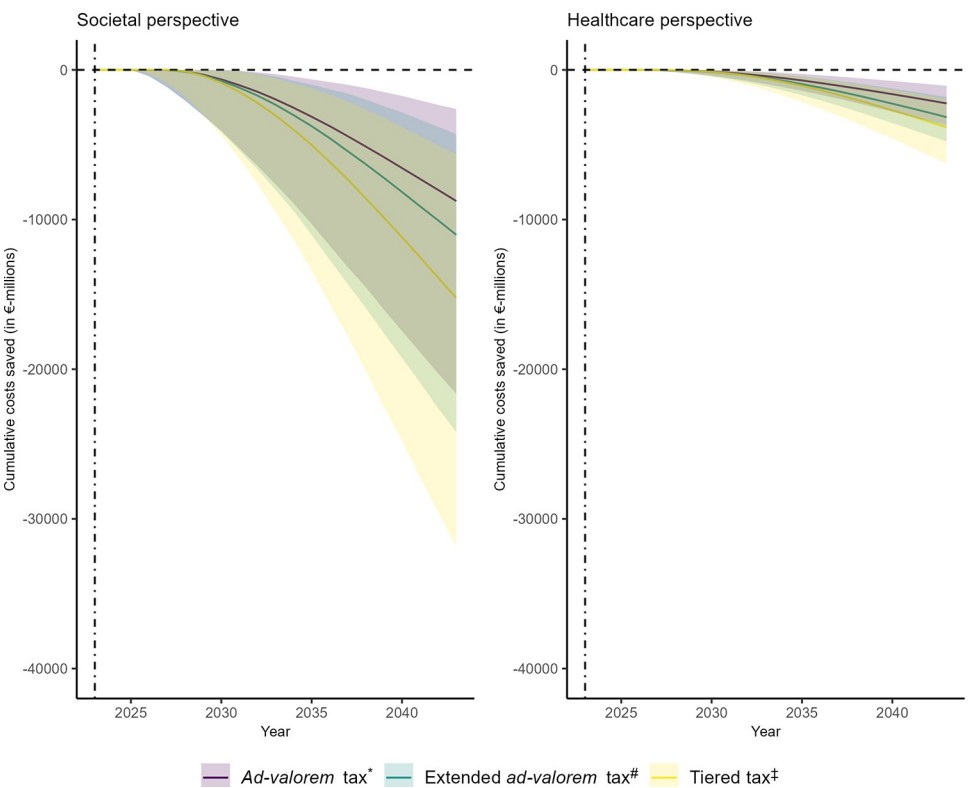

**Fig 3. Cumulative costs saved through SSB taxation from 2023 to 2043 by scenario and health economic perspective.** Line plots of median cumulative costs saved in €-millions from 2023 to 2043 as a consequence of SSB taxation scenarios stratified health economic perspective. Shaded areas indicate 95%-uncertainty intervals. *"Ad valorem tax" refers to a 20% ad valorem tax on SSBs with a pass-through to consumers of 82% (for details, see section "Sugar-sweetened beverage taxation scenarios"). #"Extended ad valorem tax" refers to a 20% ad valorem tax on SSBs and fruit juice with a pass-through to consumers of 82% (for details, see section "Sugar-sweetened beverage taxation scenarios"). ‡"Tiered tax" refers to a tiered tax on SSBs similar to the UK SDIL that leads to a reduction in SSB sugar content by 30% through reformulation (for details, see section "Sugar-sweetened beverage taxation scenarios"). SSB, sugar-sweetened beverages; UK SDIL, United Kingdom Soft Drinks Industry Levy.

relative magnitude between policy scenarios and distribution according to sex and age (Table 2, Tables W and X in S1 Appendix). All 3 policy scenarios were cost-saving and dominant from healthcare and societal perspectives compared to their counterfactual without SSB taxation (Fig 2). Cumulative cost savings over time per scenario and perspective are shown in Fig 3. The total QALYs gained were 106,000 (95%-UI [57,200, 107,100]) for the "ad valorem tax," 252,400 (95%-UI [176,700, 325,800]) for the "extended ad valorem tax," and 192,300 (95%-UI [130,100, 254,200]) for the "tiered tax." We estimated that from a healthcare perspective €2,262 million (95%-UI [€1,189, €3,596]) costs could be saved with the "ad valorem tax"; €3,141 million (95%-UI [1,942, 4,568]) with the "extended ad valorem tax"; and €3,850 million (95%-UI [€2,070, €6,075]) with the "tiered tax." The largest share of disease-specific costs saved in all scenarios was due to T2DM (Table 2). From a societal perspective, economic gains were many times bigger (€9,584 million, 95%-UI [€4,714, €15,304] savings for the "ad valorem tax"; €11,827 million, 95%-UI [€6,702, €17,887] for the "extended ad valorem tax"; €16,013 million, 95%-UI [€8,090, €25,500] for the "tiered tax"). Productivity gains were largely determined by the prevention of premature deaths and T2DM-related sick leave and time costs (Table 2).

## Validation results

IMPACT$_{NCD}$ performed very well in internal and external validation analyses (Fig S–AM in S1 Appendix). In the cross-validation with PRIMEtime, we found that both models estimated similar health impacts, supporting the overall robustness of our findings. Uncertainty in IMPACT$_{NCD}$ was generally higher than in PRIMEtime but for all scenarios and outcomes, uncertainty intervals overlapped (Fig 4, Fig AN and AO in S1 Appendix).

## Structural uncertainty and sensitivity analyses

When excluding direct, BMI-independent effects of SSBs, the health and economic impact across all scenarios, particularly on T2DM and CHD, was considerably smaller (Fig 2, Table Y in S1 Appendix). Here, the largest health gains would be achieved in the "extended ad valorem tax" scenario (177,600 QALYs gained, 95%-UI [121,500, 242,300]; €2,704 million costs saved from societal perspective, 95%-UI [€1,345, €5,002]). However, considering only SSBs, the "tiered tax" would on average produce larger benefits (73,500 QALYs gained, 95%-UI [51,600,

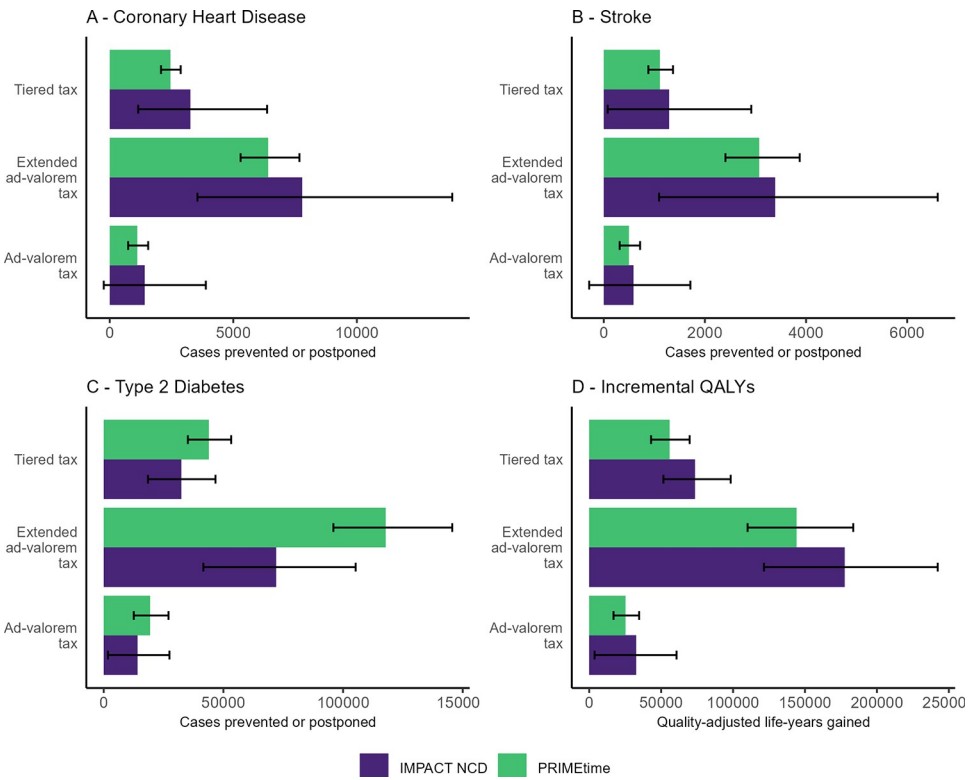

**Fig 4. Cross-validation results of IMPACT$_{NCD}$ Germany and PRIMEtime with key health outcomes.** Horizontal bar chart comparing cross-validation outcomes for (A) CHD cases prevented/postponed, (B) stroke cases prevented/postponed, (C) type 2 diabetes cases prevented/postponed, and (D) QALYs gained between the IMPACT$_{NCD}$ microsimulation (purple) and the PRIMEtime cohort model (green). Only BMI-mediated exposure pathways are modelled. Error bars indicate 95%-uncertainty intervals. "Ad valorem tax" refers to a 20% ad valorem tax on SSBs with a pass-through to consumers of 82% (for details, see section "Sugar-sweetened beverage taxation scenarios"). "Extended ad valorem tax" refers to a 20% ad valorem tax on SSBs and fruit juice with a pass-through to consumers of 82% (for details, see section "Sugar-sweetened beverage taxation scenarios"). "Tiered tax" refers to a tiered tax on SSBs similar to the UK SDIL that leads to a reduction in SSB sugar content by 30% through reformulation (for details, see section "Sugar-sweetened beverage taxation scenarios"). BMI, body mass index; NCD, noncommunicable disease; QALY, quality-adjusted life year; SSB, sugar-sweetened beverages; UK SDIL, United Kingdom Soft Drinks Industry Levy.

98,400]; €1,292 million costs saved from societal perspective, 95%-UI [€620, €2,292]) than estimated in the "ad valorem tax" scenario (32,600 QALYs gained, 95%-UI [3,700, 60,700]; €589 million costs saved from societal perspective, 95%-UI [€128, €1,352]). A detailed comparison for all health and economic outcomes is available in Table Y in S1 Appendix.

Health and economic impacts changed sometimes significantly when varying the tax rate for the "ad valorem tax" scenario between 10% (change in QALYs: −43%) and 30% (+48%), excluding substitution effects from SSBs to fruit juice (+6%), or implementing the effect of taxation using a meta-analytic estimate (−25%); decreasing reformulation for the "tiered tax" to 10% (−66%); or varying the discount rate for costs and QALYs (−66% to +30% depending on the discount rate and scenario). We estimated that observed voluntary industry commitments to reduce sugar content in SSBs would only gain 16,400 QALYs and that a combined reformulation and consumption reduction of SSBs would lead with 322,200 QALYs gained to the highest impact (Tables T–V in S1 Appendix).

## 4. Discussion

We use established epidemiological evidence and national data to develop the first population health microsimulation model for Germany (IMPACT$_{NCD}$), which we apply to estimate the health and economic impact of 3 SSB taxation scenarios. Our study suggests that the implementation of a German national ad valorem or tiered tax on SSBs following international scientific recommendations and examples such as Mexico and the UK and considering the additional taxation of fruit juice, could lead to substantial health gains (approximately 106,000 to approximately 250,000 QALYs gained) and save a substantial amount of costs from both healthcare (approximately €2,300 to approximately €3,800 million) and societal perspectives (approximately €9,600 to approximately €16,000 million) over the next 20 years. We find that a reduction of SSB sugar content, as observed following the introduction of the UK SDIL, is likely to lead to higher health gains and averted costs than the consumption reduction that could be expected from a 20% ad valorem tax without any reformulation. We show that the additional taxation of fruit juice would lead to increased health gains in the "ad valorem tax" scenario and could be justified due to high consumption levels in Germany. In sensitivity analyses, we find that health impacts that can be expected from observed voluntary industry commitments to reduce sugar in SSBs are negligible compared to the modelled SSB taxation scenarios. We also estimate that the health and economic impact of SSB taxation is considerably higher, if potential direct (BMI-independent) effects of sugar in SSBs are confirmed but that a substantial policy impact can still be expected if only established BMI-mediated effects on cardiometabolic outcomes are considered. Results from validation analyses encourage trust in the model which cross-validated well with the independently developed PRIMEtime cohort model.

Our study is consistent with international modelling studies that have shown that SSB taxation may lead to substantial long-term health and economic impacts, primarily through a reduction in BMI due to lower sugar consumption and the prevention of T2DM [28,77–79]. However, we explicitly show that the quantification of the SSB taxation impact depends to a large degree on the relevance of direct, BMI-independent health effects of SSBs from nutritional epidemiological studies which few modelling studies have considered [28,30]. By transparently reporting the resulting uncertainty, we provide a corridor for the predicted policy impact and highlight the need for robust causal epidemiological estimates. To the best of our knowledge, our study is the first diet policy evaluation to compare individual-level and cohort modelling approaches and with 2 separate, independently developed simulation models. Similar efforts have so far only been made for diabetes- and cancer-specific, clinically oriented

models but are very important to improve the quality and trustworthiness of modelling studies [80,81].

We add to the few previous simulation-based studies which have analysed the potential impact of SSB taxation in Germany by providing a much more comprehensive and detailed analysis of different policy scenarios based on real-world examples [16–18]. While we come to the same overall conclusion that taxation of SSBs is very likely to have positive impacts on population health, we can provide concrete policy guidance that a tax design focusing on SSB sugar content is likely most effective. However, this depends to some degree on the relevance of BMI-independent health effects of SSBs. We also quantify the positive cardiometabolic health effects of SSB taxation in Germany and the substantial societal economic impact from productivity losses and patient time costs for T2DM self-management which is more than twice as high as the averted medical costs.

Our findings are supported by evidence on the increase of SSB prices under volumetric or ad valorem taxation, corresponding reductions in SSB purchases and observed reformulation under tiered taxation [4,32,48,82–84]. Robust evidence from randomised trials and cohort studies shows that increased sugar and SSB consumption leads to weight gain [3,85,86]. Recently, observational evidence on the effect of implemented SSB taxes on anthropometric outcomes is emerging [8].

A particular strength of our approach is the use of an established modelling framework, which enables the detailed, flexible simulation of the health of the German population under different scenarios. To our knowledge, we are the first to model the full health and economic impact of SSB taxation on cardiometabolic health in Germany based on international scientific recommendations and implemented policies. Further, our model validates very well, including on externally observed data, and we adhere to transparency guidelines by making all code available in a public repository. We comprehensively consider implications of all sources of uncertainty from parameter uncertainty to the included risk relationships and the chosen simulation method, which has not been done before in population health modelling. This makes our findings particularly robust. Finally, we provide concrete policy recommendations for the new German national strategy on food and beyond.

However, several limitations need to be considered. First, Germany lacks a regular health examination survey that includes granular, individual-level dietary intake data, and the last official national nutrition survey is from 2007 [12]. We therefore had to rely primarily on the KORA S4 cohort study with its 2 follow-ups F4 and FF4 (1999 to 2014) for anthropometric and nutrition data that was designed to be population representative of the Augsburg region in southern Germany [36]. Thus, full representation of the German population with regards to the modelled exposures might be biased and should be addressed in future iterations of the model if appropriate data sources are available. Second, we were not able to include time trends in beverage consumption which was only collected in FF4. A comparison with annual, industry-reported, aggregated beverage consumption data indicates that (1) SSB and juice intake in Germany has remained stable; (2) SSB consumption is likely underreported; and (3) juice consumption may be overreported (Methods A in S1 Appendix under "Exposure module"). Third, we do not account for health effects beyond CVD and T2DM. Fourth, we do not consider cumulative effects of sugar intake over the life course. Fifth, we do not incorporate heterogeneity in price elasticities by age, sex, or other characteristics. Additionally, our elasticities were estimated with data prior to the recent rise in inflation. However, according to economic theory, price elasticities should generally be higher during inflation as real income diminishes. Sixth, we did not include health effects in children or adolescents. Importantly, the above limitations are unlikely to meaningfully impact our results, but indeed most likely lead to underestimation of the policy impact. Seventh, we cannot perform a detailed simulation of

SSB taxes depending on the sugar content of certain products, such as the SDIL, because we do not have access to product-level data. This also does not allow us to perform a standardised comparison of different tax designs or to distinguish tax-related price and reformulation effects in the tiered tax scenario. The interpretation of the relative effectiveness of the simulated tax designs should be made with these limitations in mind. Similarly, in the "ad valorem tax" scenario, we approximate the effects of volumetric taxes which were most often designed to increase prices by 10% to 20% [48]. Eighth, we have neither included costs of implementing an SSB tax, nor industry reformulation costs. Because no good guidance on SSB tax implementation cost exists, previous studies have for example assumed that these would be around 2% of the tax payments [87] or general administrative and auditing costs [88]. However, while implementation costs might be higher for tiered taxes due to their more complex design, considering the estimated population-level economic impacts these are negligible. Finally, as with all population health modelling studies, our results are subject to the validity of the underlying economic and epidemiological evidence including the employed estimates of long-term weight change. We account for the arising epistemic uncertainty by relying on high-quality risk estimates from decades of research and by incorporating as many sources of uncertainty in our simulations as possible.

Our findings have implications for German health and fiscal policy makers and the new national food strategy. We provide evidence that SSB taxation is a cost-effective tool to address the burden of NCDs in Germany. While we acknowledge that voluntary industry commitments to reduce SSB sugar content are in place, a previous analysis has shown that they fail to achieve targeted reductions [15]. We demonstrate that SSB taxation would lead to approximately 10 times larger healthcare cost savings compared to these voluntary commitments. We particularly show that focusing on reductions of SSB sugar content, for example, through a tax design incentivizing reformulation like the UK SDIL, might have larger health and economic impacts than ad valorem taxation of SSBs with moderate tax rates. This finding is consistent with a study from the US [28]. Achieving both a reduction in sugar content of SSBs and their consumption would be most effective. Considering that government efforts to reduce sugar content in SSBs through voluntary commitments were unsuccessful, the introduction of a tiered tax levied on producers, taking the UK SDIL as a blueprint, which gives producers the option to avoid the tax, seems most feasible in the German context.

Recent data from the UK government shows that the tax revenue from the SDIL accumulates to more than £300 million per year even after the observed product reformulation [89]. While earmarking of tax revenues to specific purposes is not possible in Germany, these resources could be acknowledged in fiscal negotiations about social and healthcare budgets for NCD preventive measures, ideally striving to alleviate existing health disparities in disadvantaged groups [2]. While we do not calculate the projected tax revenue in our study, it is important to note that different SSB tax designs have implications for this potentially relevant outcome. Since tiered taxes are designed to incentivize reformulation (i.e., to avoid the tax), their revenue might be lower than for ad valorem or volumetric taxes [90].

Based on our findings and from a public health perspective, the additional taxation of fruit juice in Germany could be justified as well, considering high consumption levels [91,92]. In fact, we predict the largest reduction in sugar consumption under the "extended ad valorem tax" (including fruit juice). Yet, this would not translate to the largest health gains if BMI-independent, direct effects of SSBs are considered. However, considering international experiences, the taxation of fruit juice is unlikely [7].

To curb the health, economic, and environmental burden of unhealthy diets over the life course, a comprehensive, population-based multicomponent policy approach transforming the food system is needed [2,93]. This is key to enable citizens to live a healthy life and protect

particularly children and adolescents from negative health consequences in adulthood. Further, the German population is progressively ageing and population-based policies which prevent NCDs, such as T2DM, which increase the likelihood for early retirement, can contribute to sustain macroeconomic productivity.

Our study supports the rationale that fiscal policies (i.e., health taxes) which address specific nutritional components or food groups should be considered key preventive policies similarly to those for tobacco control [5,94]. As such they can address negative health externalities resulting from diet-related NCDs [95]. Additionally, negative effects on employment and industry have thus far not been observed [96,97]. Structural health policies such as SSB taxation are also more likely to improve health equity overall compared to alternative, often high-agency, policies like information campaigns [98]. However, one caveat of fiscal policies is that they might be regressive [95].

To further improve the accuracy of population health modelling studies and their relevance for policy makers generally and specifically in Germany many avenues for future research exist. First, the quantification of causal effects of the cross-sectoral, multidimensional, and heterogenous behavioural response to health and social policies is of particular importance [9]. This will both, improve our understanding of policy mechanisms and lead to better projections from policy simulations. Second, while simulation models are most useful for comparing potential future policy scenarios, they can also be useful for monitoring the impact of observed exposure changes within a holistic policy evaluation framework. Here, quasi-experimental methods can be used in conjunction with novel data sources to estimate causal short- and mid-term effects of policies, while simulation models use these estimates to project long-term implications which are infeasible to observe due to identification problems [9,99]. Third, future studies should also aim to find ways of comparing outcomes from real-world evaluations of NCD policies with simulation modelling studies to improve long-term predictions [9]. Fourth, more evidence on established and potential dietary risk factors, such as artificially sweetened beverages, is needed, which ideally provides causal estimates and takes food matrix effects into account [2]. A better understanding of dietary risks, obesity incidence, and health will also lead to better NCD policy modelling studies. Finally, in Germany a better surveillance of dietary and metabolic risk factors across population strata is key to enable the identification of vulnerable population subgroups and assess the equity impact of policies. Addressing these gaps will enable better, timely policy recommendations and ideally improve population health outcomes.

In conclusion, the introduction of a 20% ad valorem or tiered SSB tax in Germany, based on scientific recommendations and taxes implemented in countries like Mexico or the UK, could help to reduce the national burden of cardiometabolic NCDs and save a substantial amount of healthcare and productivity costs. A tiered tax designed to incentivize reformulation of SSBs towards less sugar might have a larger population-level health and economic impact than an ad valorem tax that incentivizes consumer behaviour change only through increased prices.

## Supporting information

**S1 Appendix. Supplemental methods and results.** Detailed methodological description of the applied simulation models, including all preparatory analyses, model parameters and data sources. Additionally, extensive results tables and figures for all sensitivity analyses. (DOCX)

**S1 File. Consolidated Health Economic Evaluation Reporting Standards (CHEERS) 2022 statement.** (DOCX)

## Author Contributions

**Conceptualization:** Karl M. F. Emmert-Fees, Ben Amies-Cull, Nina Wawro, Jakob Linseisen, Matthias Staudigel, Martin O'Flaherty, Peter Scarborough, Chris Kypridemos, Michael Laxy.

**Data curation:** Karl M. F. Emmert-Fees, Nina Wawro, Jakob Linseisen, Chris Kypridemos.

**Formal analysis:** Karl M. F. Emmert-Fees, Ben Amies-Cull, Matthias Staudigel, Linda J. Cobiac, Chris Kypridemos.

**Funding acquisition:** Michael Laxy.

**Investigation:** Karl M. F. Emmert-Fees, Ben Amies-Cull, Nina Wawro, Jakob Linseisen, Matthias Staudigel, Annette Peters, Linda J. Cobiac, Peter Scarborough, Chris Kypridemos, Michael Laxy.

**Methodology:** Karl M. F. Emmert-Fees, Ben Amies-Cull, Nina Wawro, Jakob Linseisen, Matthias Staudigel, Linda J. Cobiac, Martin O'Flaherty, Peter Scarborough, Chris Kypridemos.

**Project administration:** Annette Peters, Michael Laxy.

**Resources:** Karl M. F. Emmert-Fees, Nina Wawro, Jakob Linseisen, Annette Peters, Linda J. Cobiac, Martin O'Flaherty, Peter Scarborough, Chris Kypridemos, Michael Laxy.

**Software:** Karl M. F. Emmert-Fees, Ben Amies-Cull, Matthias Staudigel, Linda J. Cobiac, Martin O'Flaherty, Peter Scarborough, Chris Kypridemos.

**Supervision:** Jakob Linseisen, Annette Peters, Linda J. Cobiac, Martin O'Flaherty, Peter Scarborough, Chris Kypridemos, Michael Laxy.

**Validation:** Karl M. F. Emmert-Fees, Ben Amies-Cull, Nina Wawro, Jakob Linseisen, Matthias Staudigel, Linda J. Cobiac, Martin O'Flaherty, Peter Scarborough, Chris Kypridemos, Michael Laxy.

**Visualization:** Karl M. F. Emmert-Fees.

**Writing – original draft:** Karl M. F. Emmert-Fees.

**Writing – review & editing:** Karl M. F. Emmert-Fees, Ben Amies-Cull, Nina Wawro, Jakob Linseisen, Matthias Staudigel, Annette Peters, Linda J. Cobiac, Martin O'Flaherty, Peter Scarborough, Chris Kypridemos, Michael Laxy.

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
