## [Editor Report · Decision Letter 0]

1 Jun 2023

Dear Dr Emmert-Fees, 

Thank you for submitting your manuscript entitled "The health and economic impacts of sugar-sweetened beverage taxation in Germany: a cross-validation modelling study" for consideration by PLOS Medicine.

Your manuscript has now been evaluated by the PLOS Medicine editorial staff as well as by an academic editor with relevant expertise and I am writing to let you know that we would like to send your submission out for external peer review.

Please re-submit your manuscript within two working days, i.e. by Jun 05 2023 11:59PM.

Kind regards,

Alexandra Schaefer, PhD

Associate Editor

PLOS Medicine

---

## [Decision Letter · Decision Letter 1]

30 Jun 2023

Dear Dr. Emmert-Fees,

Thank you very much for submitting your manuscript "The health and economic impacts of sugar-sweetened beverage taxation in Germany: a cross-validation modelling study" (PMEDICINE-D-23-01506R1) for consideration at PLOS Medicine. 

Your paper was evaluated by an associate editor and discussed among all the editors here. It was also discussed with an academic editor with relevant expertise, and sent to independent reviewers, including a statistical reviewer. The reviews are appended at the bottom of this email and any accompanying reviewer attachments can be seen via the link below:

[LINK]

In light of these reviews, I am afraid that we will not be able to accept the manuscript for publication in the journal in its current form, but we would like to consider a revised version that addresses the reviewers' and editors' comments. Obviously we cannot make any decision about publication until we have seen the revised manuscript and your response, and we plan to seek re-review by one or more of the reviewers. 

We expect to receive your revised manuscript by Jul 21 2023 11:59PM. Please email us (plosmedicine@plos.org) if you have any questions or concerns.

We look forward to receiving your revised manuscript. 

Sincerely,

Alexandra Schaefer, PhD

PLOS Medicine

plosmedicine.org

GENERAL COMMENTS

Please respond to all editor and reviewer comments.

Please cite your Supporting Information as outlined here: https://journals.plos.org/plosmedicine/s/supporting-information

Please ensure to be consistent in the use of the term ‘confidence intervals’ or ‘uncertainty intervals’.

Please include the completed CHEERS checklist as Supporting Information. When completing the checklist, please use section and paragraph numbers, rather than page numbers. 

ACADEMIC EDITOR COMMENTS

I focused on the comments of Reviewer 2. I think they are fair. They might require an appendix be added plus in the text explain the key methods points that are missing. It is very hard to evaluate the results of the simulation without understanding these 3 effects.

EDITOR-IN-CHIEF COMMENTS

Please ensure that you sufficiently address the price elasticity of SSBs in the main text. In addition, please clearly address the choice of Germany for your modelling study and discuss the representation of the German population in your model, as well as which aspects of economic, policy, and/or health care related issues may be unique to Germany. Please clarify how country-specific characteristics were captured by your model and expand your introduction/discussion on the current political/policy background on SSBs in Germany.

FINANCIAL DISCLOSURE

The funding statement should include: specific grant numbers, initials of authors who received each award, URLs to sponsors’ websites. Also, please state whether any sponsors or funders (other than the named authors) played any role in study design, data collection and analysis, the decision to publish, or preparation of the manuscript. If they had no role in the research, include this sentence: “The funders had no role in study design, data collection and analysis, decision to publish, or preparation of the manuscript.”

DATA AVAILABILITY STATEMENT

PLOS Medicine requires that the de-identified data underlying the specific results in a published article be made available, without restrictions on access, in a public repository or as Supporting Information at the time of article publication, provided it is legal and ethical to do so. Please see the policy at http://journals.plos.org/plosmedicine/s/data-availability and FAQs at http://journals.plos.org/plosmedicine/s/data-availability#loc-faqs-for-data-policy 

The Data Availability Statement (DAS) requires revision. For each data source used in your study: 

ABSTRACT

PLOS Medicine requests that main results are quantified with 95% CIs as well as p values. Please include. When reporting p values please report as p<0.001 and where higher as the exact p value p=0.002, for example. For the purposes of transparent data reporting, if not including the aforementioned please clearly state the reasons why not.

Please include any important dependent variables that are adjusted for in the analyses.

Throughout, suggest reporting statistical information as follows to improve clarity for the reader “22% (95% CI [13%,28%]; p</=)”. Please amend throughout the abstract and main manuscript.

Please note the use of commas to separate upper and lower bounds, as opposed to hyphens as these can be confused with reporting of negative values.

When a p value is given, please specify the statistical test used to determine it. Please report p values as p<0.001 and where higher as 'p=0.002'.

Please ensure that all numbers presented in the abstract are present and identical to numbers presented in the main manuscript text.

l.42: Please define ‘UK’.

l.60: Please change ‘find’ to ‘found’ and ensure to use the same tense for presenting results.

l.62-66: For clarity, please add details about the ad-valorem tax and the tiered tax to the numbers presented in your abstract.

l.71: Please define ‘NCDs’.

Abstract Methods and Findings:

* Please include the study design, population and setting, number of participants, years during which the study took place, length of follow up, and main outcome measures.

Abstract Conclusions:

*Please begin your Abstract Conclusions with "In this study, we observed ..." or similar, to summarize the main findings from your study, without overstating your conclusions. Please emphasize what is new and address the implications of your study, being careful to avoid assertions of primacy. 

AUTHOR SUMMARY

At this stage, we ask that you include a short, non-technical Author Summary of your research to make findings accessible to a wide audience that includes both scientists and non-scientists. The Author Summary should immediately follow the Abstract in your revised manuscript. This text is subject to editorial change and should be distinct from the scientific abstract. Please see our author guidelines for more information: https://journals.plos.org/plosmedicine/s/revising-your-manuscript#loc-author-summary.

The summary should include 2-3 single sentence, individual bullet points under each of the questions.

It may be helpful to review currently published articles for examples which can be found on our website here https://journals.plos.org/plosmedicine/

INTRODUCTION

l.76-77 suggest: “In Central Europe, about 27% of premature deaths in 2017 were associated with dietary risk factors.”

l.90: Please change ‘indicates’ to ’indicated’.

l.107: Please define ‘UK’.

l.113: Please define ‘WHO’ at first use (l.87).

l.118: Please define ‘BMI’.

Please conclude the Introduction with a clear description of the study question or hypothesis.

Please temper assertions of primacy ("We address this gap by developing the first population health microsimulation model […]”) by adding ‘to the best of our knowledge’ or similar.

METHODS AND RESULTS

For sensitivity analyses, please present and justify the choice of method. (Based on Geoffrey P Garnett, Simon Cousens, Timothy B Hallett, Richard Steketee, Neff Walker. Mathematical models in the evaluation of health programmes. (2011) Lancet DOI:10.1016/S0140-6736(10)61505-X.)

Please discuss the scientific rationale for this choice of model structure and identify points where this choice could influence conclusions drawn. Please also describe the strength of the scientific basis underlying the key model assumptions. (Based on Geoffrey P Garnett, Simon Cousens, Timothy B Hallett, Richard Steketee, Neff Walker. Mathematical models in the evaluation of health programmes. (2011) Lancet DOI:10.1016/S0140-6736(10)61505-X.)

PLOS Medicine requests that main results are quantified with 95% CIs as well as p values. Please include. When reporting p values please report as p<0.001 and where higher as the exact p value p=0.002, for example. For the purposes of transparent data reporting, if not including the aforementioned please clearly state the reasons why not.

Please include any important dependent variables that are adjusted for in the analyses.

Suggest reporting statistical information as detailed above – see under ABSTRACT

Please present numerators and denominators for percentages, at least in the Tables [not necessarily each time they're mentioned].

Please ensure to use a consistent tense for presenting results. For example, section 3.3. is written in past tense, whereas the introduction part of the results (3. Results) is written in present tense.

l.150: Please define ‘US’.

l.221. Please change ‘consider’ to ‘considered’.

p.46: Please change “In IMPACTNCD Germany The calculation of […]” to “In IMPACTNCD Germany, the calculation of […]”.

DISCUSSION

Please present and organize the Discussion as follows: a short, clear summary of the article's findings; what the study adds to existing research and where and why the results may differ from previous research; strengths and limitations of the study; implications and next steps for research, clinical practice, and/or public policy; one-paragraph conclusion (Please remove the heading “5. Conclusion).

Please temper assertions of primacy ("We use the best available epidemiological evidence […]”; “However, we are the first to show […]”, “Our study is the first diet policy evaluation […]”. “We are also the first to quantify the positive cardiometabolic health effects of SSB taxation in […]”. “Our study is the first to model the full health […]”) by adding ‘to the best of our knowledge’ or similar.

l.540-l-545: Please clarify and explain the Copyright statement mentioned under ‘Declarations’.

p. 23-25: Please remove the Declarations section from your main manuscript. The data should only be included in the corresponding section in the online submission form. The ethical approval details should be included in the Methods section of your manuscript and in the online submission form.

FIGURES

For all Figures, please ensure that you have complied with our figures requirements http://journals.plos.org/plosmedicine/s/figures.

Please consider avoiding the use of red and green in order to make your figure more accessible to those with colour blindness.

For Figures, if applicable, please show the axis beginning at zero. If this is not possible, please show a break in the axis.

Please in the figure legend/description, define abbreviations used in each figure (including those in Supporting Information files).

Figure 1: Please define ‘∆’ in the figure description.

Figure 2: Please define ‘SSB’.

Figure 2: Please check the statement “Horizontal and vertical error bars indicate 95%-confidence intervals of incremental QALYs and costs per scenario, respectively.”

Figure 2: On the y-axis, the unit used is in “€-billions”, whereas in the main manuscript you refer to “millions”. Please revise.

Figure 3: Please define ‘SSB’.

Figure 4: Please define the meaning of the three SSB taxation scenarios in the figure description or clarify the meaning of ‘extended’ on the y-axis.

Please add figure descriptions to all supplementary figures.

Supplemental Figure 2: Please define ‘GAMLSS’ and ‘KORA’.

Supplemental Figure 4: Please explain the meaning of all lines presented in the figure.

Supplemental Figure 4: For clarity and to avoid confusion due to color, we suggest changing the figure legend to “Source; Observed – dashed line; Modelled – solid line”.

Supplemental Figure 4: Please define ‘ml’, ‘GAMLSS’, ‘KORA’, ‘NVS II’.

Supplemental Figure 5: Please define ‘GAMLSS’ and ‘ml’. Please add ‘Age, years’ as a description/unit to the age groups presented on the left side of the figure.

Supplemental Figure 6: Please define ‘GAMLSS’ and ‘ml’. 

Supplemental Figure 7: Please define ‘GAMLSS’.

Supplemental Figure 8: Please explain the meaning of all lines presented in the figure.

Supplemental Figure 8: For clarity and to avoid confusion due to color, we suggest changing the figure legend to “Source; Observed – dashed line; Modelled – solid line”.

Supplemental Figure 8: Please define ‘ml’, ‘GAMLSS’, ‘KORA’, ‘NVS II’.

Supplemental Figure 9: Please define ‘GAMLSS’ and ‘ml’. Please add ‘Age, years’ as a description/unit to the age groups presented on the left side of the figure.

Supplemental Figure 10: Please define ‘GAMLSS’ and ‘ml’. 

Supplemental Figure 11: Please define ‘GAMLSS’ and ‘ml’. 

Supplemental Figure 11: Please add a unit (years) for the age groups and change the description from “(blue dashed lines)” to “(blue lines)”.

Supplemental Figure 12: Please define ‘GAMLSS’ and ‘ml’. 

Supplemental Figure 12: Please change the description from “(blue dashed lines)” to “(blue lines)”.

Supplemental Figure 13: Please define ‘GAMLSS’ and ‘ml’. 

Supplemental Figure 13: Please add a unit (years) for the age groups and change the description from “(blue dashed lines)” to “(blue lines)”.

Supplemental Figure 14: Please define ‘GAMLSS’ and ‘ml’. 

Supplemental Figure 14: Please change the description from “(blue dashed lines)” to “(blue lines)”.

Supplemental Figure 15: Please define the meaning of the black dashed line and add a unit for age.

Supplemental Figure 16: Please define ‘∆’ in the figure description and change “BMI-independetn healthe effects” to “BMI-independent health effects”.

Supplemental Figure 17: Please define ‘KORA’ and add a unit for age. Please define the meaning of the dots and lines in the graphs.

Supplemental Figure 18: Please ensure to add the reason(s) for record exclusion.

Supplemental Figure 19/20/21/22/23/24: Please add a unit for age (age groups).

Supplemental Figure 25/26/27/28/29/30: Please change groups from ‘Women’ and ‘Men’ to ‘Female’ and ‘Male’. Please add a unit for age and add ‘rate’ on the y-axis (incidence/prevalence rate).

Supplemental Figure 31/32/34/35/38/39: Please add a unit for age. 

Supplemental Figure 33: For clarity, please add the years to the x-axis on the upper graph (KV 2013-2018). Please add a unit for age.

Supplemental Figure 34: Please add a unit for age.

Supplemental Figure 36/37: Please add a unit for age. Please define ‘AOK’.

Supplemental Figure 41: Please change groups from ‘Women’ and ‘Men’ to ‘Female’ and ‘Male’.

Supplemental Figure 42/43: Please change groups from ‘Women’ and ‘Men’ to ‘Female’ and ‘Male’.

Supplemental Methods 2: Please define ‘NCD’, ‘BMI’, ‘CHD’, ‘QALY’.

Supplemental Methods 3: Please define ‘CHD’, ‘QALY’.

TABLES

Please note the use of commas to separate upper and lower bounds, as opposed to hyphens as these can be confused with reporting of negative values. Suggest reporting statistical information as detailed above – see under ABSTRACT

Please define all abbreviations used in the table below each table (including those in Supporting Information files).

For clarity, please add the exact currency (€) to cost-related numbers (€-millions). Please revise throughout your entire manuscript.

Table 1/2: In your table legend, please define the meaning of the three SSB taxation scenarios (at minimum, please define ‘20% extended ad valorem tax’).

Supplemental Table 1: Please define all abbreviations used in the table (e.g., KORA, NVS II) below the table.

Supplemental Table 3: Please define ‘SSB’.

Supplemental Table 4: Please define ‘KORA’, ‘NVS II’ and ‘n/a’.

Supplemental Table 5: Please define ‘KORA’, ‘NVS II’ and ‘n/a’.

Supplemental Table 6: Please define ‘Y’.

Supplemental Table 7: Please define ‘n/a’ and add a unit for age.

Supplemental Table 12: Please define ‘EQ-5D-5L’

Supplemental Table 15b: Please define ‘CHD’.

Supplemental Table 16: Please add a unit for age. Please change the categories under “Sex” from ‘Women’ and ‘Men’ to ‘Female’ and ‘Male’.

Supplemental Table 17: Please add a unit for age. Please change the categories under “Sex” from ‘Women’ and ‘Men’ to ‘Female’ and ‘Male’. Please add a unit (€) to all cost-related numbers.

Supplemental Table 18: Please change the categories from ‘Women’ and ‘Men’ to ‘Female’ and ‘Male’ and add details about the age groups (category and unit). Please define ‘Juice’, ‘ml’, ‘20% extended ad valorem tax’.

Supplemental Table 19/20: For clarity, please add the exact currency (€) to cost-related numbers (€-millions). 

Supplemental Table 21: For clarity, please add the exact currency (€) to cost-related numbers (€-millions). Please define ‘20% extended ad valorem tax’.

Supplemental Table 22: Please add a unit for age and costs. Please define ‘20% extended ad valorem tax’.

Supplemental Table 23: Please add a unit for costs. Please change the categories from ‘Women’ and ‘Men’ to ‘Female’ and ‘Male’. Please define ‘20% extended ad valorem tax’. Please remove ‘ICER’ from the abbreviations. 

Supplemental Table 24: Please add a unit for costs. Please define ‘20% extended ad valorem tax’ and ‘BMI’. Please remove ‘ICER’ from the abbreviations. 

Supplemental Table 25: Please define ‘20% extended ad valorem tax’.

REFERENCES

PLOS uses the numbered citation (citation-sequence) method and first six authors, et al.

Please ensure that journal name abbreviations match those found in the National Center for Biotechnology Information (NCBI) databases (http://www.ncbi.nlm.nih.gov/nlmcatalog/journals), and are appropriately formatted and capitalised.

Please also see https://journals.plos.org/plosmedicine/s/submission-guidelines#loc-references for further details on reference formatting. 

Where website addresses are cited, please specify the date of access. 

Supplemental References: RE reference [66], Supplementary Material, listed as under review, papers cannot be listed in the reference list until they have been accepted for publication or are otherwise publicly accessible (for example, in a preprint archive). The information may be cited in the text as a personal communication with the author if the author provides written permission to be named. Alternatively, please provide a different appropriate reference.

Comments from the reviewers:

Reviewer #1: Recognizing a gap in context-specific research, the authors aimed to develop the first population health microsimulation model for Germany, termed IMPACTNCD Germany. This model assessed the potential impact of different SSB taxation on sugar consumption and thus on diseases like stroke, coronary heart disease, and type 2 diabetes, considering morbidity, mortality, healthcare costs, patient time, and productivity costs. Overall, this is a well-designed and conducted study. The methods of the study are rigorous to me, and the results are very useful to inform policies of SSB taxation in Germany. Below are my specific comments. 

1. The data are not fully publicly available, which may not meet PLoSMED's requirement. 

2. Line 90: it could be helpful to provide brief introduction of "tiered structure" in the introduction. Not all readers are familiar with this term. 

3. Line 76-120: The introduction does a great job of establishing the importance of dietary risk factors in premature deaths and the role of SSBs. However, the connection between SSBs and dietary risk factors could be better emphasized.

4. Line 76-120: While the authors point out that previous studies did not quantify healthcare cost savings or productivity costs, they could go further in explaining why their approach is novel or important. For instance, what is the specific benefit of using a microsimulation model for Germany? Why are these three policy scenarios chosen for simulation?

5. Line 184: "..the distribution of BMI.." Do you mean the distribution of SSB consumption here? The surveys you described in this paragraph were all about dietary consumption not about anthropometric measurements.

6. Line 198-199: I think the Global Burden of Disease Study (the GBD) should have better incidence estimates for stroke and CHD for Germany. 

7. Line 223-224: was this assumption based on the reference cited here? Should make it clear how the assumption is made. 

8. Line 234: what do you mean by conservative aetiologic effects here? Lower bound of the 95% CI? Need to be more specific.

9. Line 285: I'm still not sure whether discounting QALYs is a good idea.. To me A QALY should equal a QALY no matter whether that is now or 20 years from now. 

Reviewer #2: Thanks for allowing me to review the paper entitled "The health and economic impacts of sugar-sweetened beverage taxation in Germany: a cross-validation study". I want to highlight the solid specification of the epidemiological model due to the impressive validation process by the authors by calibrating this model according to observed health information. Due to the strength of this epidemiological model, most of my comments are related to the economic aspects of the SSB tax assessments. I hope the authors find these comments relevant to strengthen the study. 

Major comments

The main text does not offer enough information to understand/identify the different tax burdens depending on the sugar-content threshold under the tiered tax. For example, what is the expected price increase with/without reformulation under the tiered tax in Germany? For the tiered tax, could the authors explain how they isolated the tax-related reformulation effect from the tax-related price effect? Moreover, the current approach does not allow a "standardized" comparison between the 20% ad valorem tax and the tiered tax. Thus, it might not be possible to conclude if the higher health effects under the tiered tax are because of reformulation or more significant price increases. I recommend that the authors review the paper by Salgado and Ng (2021) where the authors make volumetric and sugar taxes comparable before reformulation so that it is possible to see the potential health effect linked to reformulation. 

Authors gather information on price elasticities from a "forthcoming" paper (see reference #45). Estimations of price elasticities are a demanding task by itself. Thus, if the relevant paper is still under review, estimates of price elasticities can change due to reviewers' comments. Thus, I recommend that the authors use published evidence either from a meta-analysis or another paper for Germany. Or is this point already covered under the sensitivity analysis of "implementing the effect of taxation using a meta-analytic estimate? If so, it is important to highlight the remarkable difference (i.e., 25%) compared to their preferred model specification.

Regarding the epidemiological model, can a cohort member of the microsimulation get two or more diseases? 

Could the authors include estimates of sugar reductions in Table 1 as percentage reductions? This information can be policy-relevant due to a direct interpretation of the tax effectiveness is in terms of percentage reductions.

There is a marked reduction (i.e., five times larger) in sugar consumption from SSB and juice when comparing scenarios 1 and 2 in Table 1. Could the authors provide some insights to understand this marked difference between scenarios? For example, is it a result of a high baseline consumption of juices in Germany? Is it a result of SSB and juices being complements? 

While Table 1 shows that the "20% extended ad valorem tax" leads to a sugar consumption reduction that more than doubles the reduction under the tiered tax, table 2 shows, in general, larger health effects under the latter tax. These results might seem counterintuitive because we can expect larger health effects under policies that lead to higher sugar reductions, which would be the "20% extended ad valorem tax" in this paper. Could the authors explain this counterintuitive result?

According to my best understanding, the authors solely focused on health savings overlooking any cost linked to the tax implementation. Other scholars have included costs. For example, see the papers by (Basto-Abreu et al., 2019; Wilde et al., 2019). Could the authors include any costs linked to the tax implementation? 

A downside for sugar taxes that encourage reformulation is a potential reduction in tax revenue (see findings by Salgado and Ng(2021)). While I do not recommend including tax revenue as part of the cost-effectiveness analyses (because it is a benefit for the government but a cost for consumers), I think it might be helpful to discuss this potential tax revenue reduction as a relevant outcome for the government when weighing benefits and costs across different tax designs. 

Minor comments

 In the abstract, the tax in Mexico corresponds to 1.00 peso/liter rather than 0.01 peso/liter.

 The authors highlight that their tax-design selection is "based on internationally implemented policies" (see line 498). While this is clear in the case of the tiered tax, which corresponds to the design in the UK, it is not obvious which country is the reference for the 20% tax. 

 It would be policy-relevant to see a clear justification regarding the selection of tax designs for main scenarios 2) and 3). For example, why did the authors prefer a sugar-tax design as in the UK rather than the sugar tax as in South Africa (which has been associated with a sugar reduction larger than 50% (Stacey et al., 2021))? 

 For Table 2, could the author confirm whether uncertainty intervals (UI) correspond to 95% UI? If so, please include this information as part of the footnote. 

Reference 

Basto-Abreu, A., Barrientos-Gutiérrez, T., Vidaña-Pérez, D., Colchero, M. A., Hernández-F, M., Hernández-Ávila, M., Ward, Z. J., Long, M. W., & Gortmaker, S. L. (2019). Cost-Effectiveness Of The Sugar-Sweetened Beverage Excise Tax In Mexico. Health Affairs (Project Hope). https://doi.org/10.1377/hlthaff.2018.05469

Salgado Hernández, J. C., & Ng, S. W. (2021). Simulating international tax designs on sugar-sweetened beverages in Mexico. PLOS ONE, 16(8), e0253748. https://doi.org/10.1371/journal.pone.0253748

Stacey, N., Edoka, I., Hofman, K., Swart, E. C., Popkin, B., & Ng, S. W. (2021). Changes in beverage purchases following the announcement and implementation of South Africa's Health Promotion Levy: an observational study. The Lancet Planetary Health, 5(4), e200-e208. https://doi.org/10.1016/S2542-5196(20)30304-1

Wilde, P., Huang, Y., Sy, S., Abrahams-Gessel, S., Jardim, T. V., Paarlberg, R., Mozaffarian, D., Micha, R., & Gaziano, T. (2019). Cost-effectiveness of a US national sugar-sweetened beverage tax with a multistakeholder approach: Who pays and who benefits. American Journal of Public Health. https://doi.org/10.2105/AJPH.2018.304803

Reviewer #3: Dear Authors

Thank you for very well written study. It is one of the best-documented study in the public health literature that I have had the opportunity to review. 

I do have a few questions for my own clarification and comments for your kind consideration. 

1. The study has modeled trend in BMI using three surveys. As I am not familiar with public policies in Germany, I would like to know whether there were any obesity/DM related campaigns in Germany during the time covered by these surveys, such that the trends might be affected by it. For example, smoking trend in absence of government push for reduction of smoking would be different as compared to a landmark report about negative consequences of smoking. Similarly if there were school-based campaigns to reduce SSB advertising and consumption in/around school-children, then the entire base of population entering the cohort-simulation model will have different BMI distribution. The external validity of the procedure to determine trends for BMI will depend on this. 

2. Inflation: Page 44 indicates that CPI-Health has been used to adjust for inflation in health-care sector. Pardon me, but I could not find information on whether the authors have adjusted the prices of SSBs and fruit-juices for inflation over the 20-year time horizon. My reasons for asking are twofold: First - in ad-valorem scenario - the absolute quantum of tax increases if there is inflation and recent macroeconomic data suggests higher food inflation due to Ukraine-Russia conflict. The quantum of tax itself in this case acts as automated brakes on consumption (as compared to specific tax which declines as proportion of price if there is rapid inflation). Second, the market structure of SSB vis-a-vis fruit juice industry is likely to affect the product-specific inflation. In my country, SSBs are manufactured/sold by firms in oligopoly tending towards duopoly structure whereas fruit-juice market is much more diverse. In face of general inflation, the ability of the dominant market players to maintain prices to gain market share is likely to happen for SSBs than for fruit juices. Hence, my question about how did authors address these diverse issues of market structure and inflation for specific products. 

3. Market structure and political feasibility. I understand the study documents the costs/benefits of 3 scenarios. As a reader I would be interested to know the political feasibility and interest in implementing (a) ad-valorem vs (c) taxes/incentives for product-reformulation. Again I am not familiar with German context and hence general audience would benefit from authors' discussion of these issues. Developing countries or governments with lower bargaining power as compared to industry would prefer (a) as it signals to the politic that action has been taken, provides stable revenue; while other countries might prefer (c). Could the authors' discuss whether (a) or (c) is more likely to be feasible in their country? 

4. Elasticity estimates. Page 93 indicates estimation of own- and cross-price elasticity estimates. Were these calculated separately by age- gender- or age-by-gender groups or did the authors calculate a single elasticity estimate? If a single elasticity estimate was calculated did the authors apply it uniformly to all age-gender groups or did they scale it different factors to take into account consumption preferences? 

Reviewer #4: Overall comments

Thank you for the opportunity to review this manuscript. This is an exceptionally thorough (due to all the Supplementary Material provided) and impressive piece of work that is an important contribution to the SSB tax academic literature. The study examines the potential impacts of an SSB tax in Germany, a country that could potentially benefit, and policymakers would also likely read this paper with great interest. There are also highly detailed methods that could benefit researchers. 

The main findings of the paper are that (1) implementing an SSB tax could have health and economic benefits in Germany, and (2) a tiered tax was modelled to be more effective (of the tax structures considered). It is a strength that the authors explored a few possible SSB tax structures, which allows for even stronger policy recommendations by comparing their relative modelled impacts. However, it should be noted there are other possible tax structures, such as the South Africa Health Promotion Levy, which taxes based on sugar concentration, and is different from the UK SDIL tiered tax.

I have few critiques as many of the questions I had were answered in the Supplementary Materials. 

Specific comments 

Abstract

Lines 67-68: If word count is not an issue, it could help the reader understand the magnitude of the difference between including/excluding fruit juice by putting the absolute or relative difference in the findings between the two approaches. The same could be done for analyses excluding direct health effects of SSBs.

Introduction 

Clear and concise Intro. Provides a strong rationale for the study and its potential use in informing policy. 

Methods

Lines 156-165: Although it's possible it is mentioned somewhere, it was not clear to me why an effort was clearly made to simulate an ad valorem tax with and without fruit juice, but fruit juice is not discussed in the context of a tiered tax. I see that the assumption made is simply a 30% reduction in sugar consumption across the board, without changing SSB intake. I still think for clarity it could be worth mentioning whether the 30% reduction does/does not include fruit drinks (with added sugar). As you've defined terms, it would seem not. Presumably if this does not include fruit drinks (with added sugar that could be reformulated), then including them along with other SSBs could lead to an even larger impact, given the impact was greatly increased in the ad valorem scenario + fruit drinks. This is very minor, but it seems worth explaining to fully understand how the tiered scenario compares to the other two. 

Lines 232-236: can you add the quantitative details for the "conservative aetiologic effect estimates" (i.e. why it is considered conservative) and "predicted reductions in weight are generally lower than using traditional energy balance equations." As an example for comparison, the above quotation from Lines 213-214 was easy to follow as it included the assumption for price pass through: "We assumed that the proportion of the SSB price increase passed on to consumers was 82% (66 to 98) based on a recent meta-analysis [4]." Some of the referencing without including quantitative details requires the reader to do a lot of tracking down of references or tables elsewhere in the document. However, I did find Supplementary Table 7 helpful in answering a lot of these questions. 

Line 266: please also provide quantitative details for "medical costs for the treatment of stroke, CHD, and T2DM."

Results

Lines 392-393: I understand the results are all available in Tables referenced, but this description ("smaller, however still large") is too vague. I would suggest adding quantitative details regarding how much smaller, or why it should still be considered large. 

Discussion 

Line 413: "(~€9,600 to ~€16,000) over the next 20 years" are you missing the word "million" as was included in line 412? It might be implied but seems clearer to include. 

Lines 486-488: In the section on future research, there are useful suggestions. Could you please add more detail about future research directions that address actual policy impacts? There is a mention of assessing equity, but Reference 89 still takes measured changes in SSB intake and then models expected changes in QALYs, which is common practice but may require real world evaluation. 

Line 527-531: "Finally, as with all population health modelling studies, our results are subject to the validity of the underlying epidemiological evidence…" Although these appear to be reliable risk estimates, using them in these types of causal models (SSB tax would lead to) is potentially pushing them to their limit. How have other modelling studies performed when compared to real world evaluations of impacts of SSB taxes on BMI and health outcomes? Although there might be decent confidence in the potential health and therefore also economic benefits, it seems conceivable that a few grams per capita per day reduction does not translate to changes in outcomes of those magnitudes, unless real impacts are measured.

[LINK]

---

## [Decision Letter · Decision Letter 2]

18 Sep 2023

Dear Dr. Emmert-Fees,

Thank you very much for re-submitting your manuscript "The health and economic impacts of sugar-sweetened beverage taxation in Germany: a cross-validation modelling study" (PMEDICINE-D-23-01506R2) for review by PLOS Medicine.

Thank you for addressing the comments from the editors and reviewers in detail. I have discussed the paper with my colleagues and the academic editor, and it has also been seen again by three of the original reviewers. The changes made to the paper were satisfactory to the reviewers, and any remaining comments are editorial in nature. We kindly request that you address the detailed editorial comments (found below). When submitting your revised paper, please include once again a detailed point-by-point response to the editorial comments.

I am pleased to say that provided the remaining editorial and production issues are dealt with we are planning to accept the paper for publication in the journal.

[LINK]

We expect to receive your revised manuscript within 1 week. Please email me (aschaefer@plos.org) if you have any questions or concerns.

If you have any questions in the meantime, please contact me (aschaefer@plos.org) or the journal staff on plosmedicine@plos.org.  

We look forward to receiving the revised manuscript by Sep 25 2023 11:59PM.   

Sincerely,

Alexandra Schaefer, PhD

Associate Editor 

PLOS Medicine

plosmedicine.org

Requests from Editors:

GENERAL COMMENTS

Thank you for considered and detailed responses to editor and reviewer comments.

Please see below for further minor points that we request you respond to in full.

1) We note that you switch between the descriptions “ad-valorem tax” and “20% ad-valorem tax”. Please note that you introduced this scenario as “ad-valorem tax” in line 252 and we suggest using a consistent format following this.

2) Please note that it is necessary that you define the numbers presented in tables and figures indicating that in parentheses you are presenting, e.g. 95%-uncertainty intervals. Please thoroughly revise your manuscript including the Supplementary material regarding this comment.

3) Please use a consistent format when using ‘Female’ and ‘Male’ as titles in figures and table. We suggest starting these words with a capitalized letter. Please revise thoroughly.

4) Any mentions of age must have a unit (e.g., years). Please revise thoroughly.

5) Please revise your manuscript thoroughly regarding the introduction of abbreviations (at first use) and when used in figures and tables.

EDITORIAL COMMENTS

We suggest changing the title to “Predicted health and economic impacts of sugar-sweetened beverage taxation in Germany: a cross-validation modelling study” and re-wording “save billions of societal costs” in the abstract.

DATA AVAILABILITY

Thank you for the updated Data Availability Statement (DAS). Please note that a study author cannot be the contact person for the data so the statement “Additional code used for data cleaning and preparation is available upon request from the corresponding author.” is not acceptable in its current form.

ABSTRACT

l.50: Please change to ‘aged 30-90 years’.

l.59: Please define ‘BMI’.

ll.65-70: Please revise this section. It reads confusingly – are there only numbers included for obesity and type 2 diabetes? Why not for cardiovascular disease? It might be clearer to insert the numbers following the disease.

AUTHOR SUMMARY

Thank you for providing the Author Summary. The Author Summary is very long and needs to be shortened. It should contain only the most important details for each of the questions, in a condensed form. For example, in the first bullet point of “Why Was This Study Done?”, we suggest removing the examples in parentheses as this goes beyond the scope of the bullet points. Similarly, the first bullet point of “What Did the Researchers Do and Find?“ could be shortened to “We evaluated three SSB taxation scenarios in Germany with a simulation model: 1) 20% ad-valorem tax on SSBs; 2) extended 20% ad-valorem tax on SSBs and fruit juice; 3) tiered tax leading to reformulation of SSBs towards 30% lower sugar content.” or similar.

It may be helpful to review currently published articles for examples which can be found on our website here https://journals.plos.org/plosmedicine/

INTRODUCTION

l.155: Please define ‘₤’.

l.156: Please define ‘g’ and ‘ml’ (at first use).

METHODS AND RESULTS

Please remove the numbering of the subheadings.

l.211: Please change to ‘age 30 to 90 years’. Please ensure to add a unit for age and revise throughout the entire manuscript.

l.245: Please change ‘Thes’ to ‘The’.

l.282: Please change to “we used data from the three KORA waves to incorporate time trends”.

l.307: Please provide a reference.

l.369: Please define ‘CVD’ (at first use).

l.380: Please change to ‘age of 65 years’.

l.435: Please define ‘PSA’.

l.478: Please note that in most instances you have written ‘20% ad-valorem tax’ in quotation marks (e.g., see l.316 “20% ad-valorem tax”). Please use a consistent format and revise throughout the entire manuscript.

ll.486-488: You state that “[…] the largest impact on obesity, particularly among women due to their relatively high fruit juice consumption (Table 2).” and reference Table 2, whereas Table 2 only shows the influence on all individuals and not grouped by sex. Please add an appropriate reference.

l.498: Please change to ‘below 70 years’.

ll.533-537: Why do you start presenting the costs in billions at this point, when you have consistently used millions before? Please keep the format consistent (we suggest millions).

DISCUSSION

l.557: Please re-phrase ‘save billions of from both healthcare […]’.

TABLES

Table 1: The table requires revision. The table is missing a description of the changes in sugar consumption being presented by sex and different age groups. The age groups lack a unit. The titles ‘Absolute change from 2023 to 2043 in g/day’ and ‘Relative change from 2023 to 2043 in %’ should be positioned in the middle of the three columns containing the actual numbers (in the current form the titles read as a title for the sex and age groups listed below). It would also be helpful to add footnotes to the different policy scenarios guiding the reader to the table description.

Table 1: The terms gender and sex are not interchangeable (as discussed in https://www.who.int/health-topics/gender); please use the appropriate term.

Table 2: We suggest adding footnotes to the different policy scenarios guiding the reader to the table description. Please define ‘Diff.’ or use the full-length word.

Table 2: We suggest changing the unit to ‘millions in €’ and removing the ‘€’ from the numbers presented in the table.

FIGURES

Please in the figure legend/description, define abbreviations used in each figure (including those in Supporting Information files).

Figure 1: Please define ‘ml’.

Figure 2: Please define ‘BMI’.

Figure 3: Please change the unit on the y-axis to ‘in €-millions’ (as in Figure 2).

Figure 3: You stated that the plots are ‘stratified by sex and health economic perspective’ while there is no stratification by sex. Please revise.

Figure 4: You have divided the four graphs into A, B, C, D and not used this in the figure description. Please revise. We suggest changing the description to “Horizontal bar chart comparing cross-validation outcomes for cases prevented/postponed for A) coronary heart disease, B) stroke, C) type 2 diabetes and D) quality adjusted life years gained between the IMPACTNCD microsimulation (purple) and the PRIMEtime cohort model (green).” or similar.

REFERENCES

Where website addresses are cited, please use ‘accessed’ instead of ‘cited’ when specifying the date of access.

Please thoroughly revise all references including those in the S1 Appendix and ensure that journal name abbreviations match those found in the National Center for Biotechnology Information (NCBI) databases (http://www.ncbi.nlm.nih.gov/nlmcatalog/journals), and are appropriately formatted and capitalised (e.g., PloS medicine should be PloS Med, BMC public health should be BMC Public Health, Diabetic medicine : a journal of the British Diabetic Association should be Diabet Med).

SUPPLEMENTARY MATERIALS

Please revise the supplementary material thoroughly regarding the definition of abbreviations (including those in figures and tables).

Please ensure that all mentions of age (including axis labels) have a unit.

Please note that column titles should include the meaning of the numbers in parentheses, e.g. Sensitivity analysis 1: Ad-valorem tax with 10% tax rate (95%-uncertainty intervals).

Supplemental Methods 1, p.6, 2nd paragraph: Please define ‘US’.

Supplemental Methods 1: p.15, 2nd paragraph: Please add a unit for age (‘aged 30 to 90 years’) and revise throughout the entire S1 Appendix.

Supplemental Figure 2: Please add an axis label for the axis of the age groups (top) and the years (right). Please define ‘kg’ and ‘m2’. We suggest changing the description to “[…] BMI distribution by 10-years age groups and year based […]” and would suggesting changing the title accordingly (like Supplemental Figure 5).

Supplemental Figure 3: Please add an axis label for the axis of the years (right). Please define ‘kg’ and ‘m2’.

Supplemental Figure 4: Please explain the meaning of the light-colored lines in the figure description (dashed and solid). Please add a unit for age.

Supplemental Figure 4: Why is there no modelled line (solid line) for the NSV II data, but a observed line (dashed line) for the combined data??

Supplemental Figure 5: Please add an axis label for the axis of the age groups.

Supplemental Figure 7: Please add a unit for age. Please define ‘KORA’.

Supplemental Figure 8: Please explain the meaning of the light-colored lines in the figure description (dashed and solid). Please add a unit for age.

Supplemental Figure 8: Why is there no modelled line for the NSV II data?

Supplemental Figure 9: Please add an axis label for the axis of the age groups. Please change the x-axis label to ‘Fruit juice consumption in ml’.

Supplemental Figure 10: Please change the x-axis label to ‘Fruit juice consumption in ml’.

Supplemental Figure 11 onward: Why did you change the data presentation from “blue dashed lines red lines” to “(solid) blue lines and red lines”? We suggest using a consistent style. 

Supplemental Figure 11/12/13/14 We suggest changing the x-axis label to ‘Sugar consumption per ml of SSB’.

Supplemental Figure 15: Please change ‘x-axes’ to ‘x-axis’. In the title, please add a unit for age (‘for ages 30 and 50 years’). 

Supplemental Table 7: Please add a unit for age in the figure description (‘of age 35 years’). For the first outcome (BMI (baseline BMI <25)), please add a unit. We suggest changing the fifth column title to include the meaning of the numbers in parentheses – are these uncertainty intervals (e.g., Risk estimate by age group* (95%-uncertainty intervals))?

Supplemental Table 8: For clarity, we suggest adding some of the information provided in Step 1 (p.33) to the table description. Please add a unit for the numbers presented in the table (years?).

Supplemental Methods 1: p.37: Please change to “Where RR_(i1 … ik) are the relative risks that are related to the specific risk exposures of the synthetic individual, as in step 1.”

Supplemental Methods 1: p.37: Please define ‘HDL’.

Supplemental Methods 1: p.38: Please write ’17,594’ instead of ‘17594’ and ensure using a consistent number format throughout the entire Appendix (e.g., also see p.56).

Supplemental Figure 17: Please consider avoiding the use of red and green in order to make your figure more accessible to those with colour blindness. We suggest adding the years of the follow-up studies to be included in the description.

Supplemental Table 9: Please add a unit for age in the figure description (‘of age 35 years’). When describing BMI, please add a unit. 

Supplemental Methods 1: p.43: Please change ‘Hessia’ to ‘Hesse’, ‘Saxonia-Anhalt’ to ‘Saxony-Anhalt’ and ‘Mecklenburg-Vorpommern’ to ‘Mecklenburg-Western Pomerania’.

Supplemental Methods 1: p.45: Please define ‘QALYs’ at first use.

Supplemental Table 12: In ‘Comments & assumptions’, please either exchange ‘Kähm et al. (2018)’ with the according reference ([33]) or add the according reference to it in the description.

Supplemental Table 13: Please explain in a footnote what the ‘EQ-5D-5L’ is.

Supplemental Methods 1: p.57: Please change ‘60.000’ to ’60,000’ and see the comment about using a consistent number format.

Supplemental Table 17/18: Please see the comment about using a consistent number format.

Supplemental Methods 1: p.61: Please change to ‘at age 60 years’.

Supplemental Figure 19: Please note that the graphs are difficult to compare due to the varying y-axis. Please ensure that the y-axis is identical for all figures to facilitate comparison. The observed mortality rates start in 2013 whereas the y-axis and black dots (presenting the observed mortality rates) start at year 1990 – please explain. From the figure description, the discrepancy between time points will not be clear to the reader. 

Supplemental Figure 20: See comment for Supp. Figure 19. Also, please change to “[…] coronary heart disease (CHD) mortality rates in Germany in women”.

Supplemental Figure 21/22/23/24: See comment for Supp. Figure 19.

Supplemental Figure 25-29: We suggest changing the x-axis label to ‘Age group (years)’. Please explain the meaning of the point clouds and the violin plot.

Supplemental Figure 25-27: Please ensure that the y-axis is identical for all figures to facilitate comparison.

Supplemental Figure 28-29: Please ensure that the y-axis is identical for all figures to facilitate comparison.

Supplemental Figure 31/32/33/34/35/36/37/38/39: Please mention in the figure description that data is presented by sex and age group.

Supplemental Figure 33: Please correct the x-axis label for the two top graphs to ‘Year’. Why does the x-axis start before 2010? Please mention in the figure description that data is presented by sex and age group. Please note that the shaded areas are hardly recognizable. 

Supplemental Figure 36: Please define ‘GEDA’.

Supplemental Methods 3: p.85: We suggest defining ‘SSB’ anew as this is a new segment of the S1 Appendix.

Supplemental Methods 3: p.86: Please define ‘BMI’.

Supplemental Figure 40/41: Please define the meaning of the three SSB taxation scenarios in the figure description. You have divided the four graphs into A, B, C, D and not used this in the figure description. Please revise (see comment Figure 4). Please define ‘CE’ in PRIMEtime CE.

Supplemental Figure 41: Where is the horizontal bar for CHD and stroke for females in the ad-valorem tax group?

Supplemental Figure 42/43: Please ensure that the y-axis is identical for figures of the same group to facilitate comparison. Please define ‘SSB’ (add in the description following ‘sugar-sweetened beverage’).

Supplemental Table 19: Please change the column title to include the meaning of the numbers in parentheses – are these uncertainty intervals (e.g., Difference to baseline in 2043 (95%-uncertainty intervals))? 

Supplemental Table 20/21: Please change the column titles to include the meaning of the numbers in parentheses – are these uncertainty intervals (e.g., Sensitivity analysis 1: Ad-valorem tax with 10% tax rate (95%-uncertainty intervals))? Please add a unit for age.

Supplemental Table 22: Please change the column titles to include the meaning of the numbers in parentheses – are these uncertainty intervals (see comment Supplemental Table 20/21)? Also, consider this for the QALYs data.

Supplemental Table 23/24/25/26: Please change the column titles to include the meaning of the numbers in parentheses – are these uncertainty intervals (see comment Supplemental Table 20/21)?

SOCIAL MEDIA

To help us extend the reach of your research, please provide any Twitter handle(s) that would be appropriate to tag, including your own, your coauthors’, your institution, funder, or lab. Please respond to this email with any handles you wish to be included when we tweet this paper.

Comments from Reviewers:

Reviewer #1: I think the authors have adequately addressed all the comments and concerns from me and other reviewers. I would recommend for acceptance. Congratulations to the authors. 

Reviewer #2: The authors have addressed all my previous comments satisfactorily. I enjoyed reviewing this paper, and congrats to the authors for their study.

Reviewer #3: The comments and revisions are satisfactory.

[LINK]

---

## [Editor Report · Decision Letter 3]

13 Oct 2023

Dear Dr Emmert-Fees, 

On behalf of my colleagues and the Academic Editor, Barry M. Popkin, I am pleased to inform you that we have agreed to publish your manuscript "Projected health and economic impacts of sugar-sweetened beverage taxation in Germany: a cross-validation modelling study" (PMEDICINE-D-23-01506R3) in PLOS Medicine.

Thank you for your very thoughtful and detailed responses to the editorial comments. We acknowledge that the revision process has been extensive, and as such, we are all the more pleased and excited to publish your manuscript. There are some remaining minor stylistic and presentation points that should be addressed prior to publication. We will carefully review whether these changes have been made. Should you have any queries or concerns regarding these final requests, please feel free to reach out to me at aschaefer@plos.org. 

Please see below for further minor points that we request you respond to:

1) In line 321, you use the abbreviation ‘CI’ for ‘confidence interval’ and introduce the abbreviation 4 lines later in line 325 (82% (95%-confidence interval [66, 98]; p<0.001)). Please revise so that the definition precedes the abbreviation.

2) In line 561, you changed “save billions of from both healthcare” to “save millions of costs from both healthcare”, even though in the revision you stated you changed it to “save a substantial amount of […]”. Please revise. We suggest changing it to “save a substantial amount of […]” in line with the Abstract.

3) Thank you for revising the Author Summary. While we feel the Author Summary improved, it is still rather long. We have adjusted and shortened the Authors Summary (please see attached file). Please feel free to use the suggested version and adjust to your liking keeping in mind to keep it at a reasonable length (similar to the one proposed by us). 

PRESS

Sincerely, 

Alexandra Schaefer, PhD 

Associate Editor 

PLOS Medicine
